# Imbalanced Mixed Linear Regression

**Pini Zilber**      **Boaz Nadler**
Faculty of Mathematics and Computer Science
Weizmann Institute of Science, Israel
`{pini.zilber, boaz.nadler}@weizmann.ac.il`

## Abstract

We consider the problem of mixed linear regression (MLR), where each observed sample belongs to one of $K$ unknown linear models. In practical applications, the mixture of the $K$ models may be imbalanced with a significantly different number of samples from each model. Unfortunately, most MLR methods do not perform well in such settings. Motivated by this practical challenge, in this work we propose `Mix-IRLS`, a novel, simple and fast algorithm for MLR with excellent performance on both balanced and imbalanced mixtures. In contrast to popular approaches that recover the $K$ models simultaneously, `Mix-IRLS` does it sequentially using tools from robust regression. Empirically, beyond imbalanced mixtures, `Mix-IRLS` succeeds in a broad range of additional settings where other methods fail, including small sample sizes, presence of outliers, and an unknown number of models $K$. Furthermore, `Mix-IRLS` outperforms competing methods on several real-world datasets, in some cases by a large margin. We complement our empirical results by deriving a recovery guarantee for `Mix-IRLS`, which highlights its advantage on imbalanced mixtures.

## 1 Introduction

In this paper, we consider a simple generalization of the linear regression problem, known as mixed linear regression (MLR) [5, Chapter 14]. In MLR, each sample belongs to one of $K$ unknown linear models, but it is not known to which one. MLR can thus be viewed as a combination of regression and clustering. Despite its simplicity, the presence of multiple linear components makes MLR highly expressive and thus a useful data representation model in various applications, including trajectory clustering [20], health care analysis [16], market segmentation [49], face recognition [7], population clustering [29], drug sensitivity prediction [37] and relating genes to disease phenotypes [8, 47].

Several methods were developed to solve MLR, including expectation maximization [15, 5], alternating minimization [52, 53] and gradient descent [56]. These methods share three common features: they all (i) require as input the number of components $K$; (ii) estimate the $K$ models simultaneously; and (iii) tend to perform better on balanced mixtures, where the number of samples from each of the $K$ models are approximately equal. As illustrated in Section 4, given data from an imbalanced mixture, these methods may fail. In addition, most of the theoretical guarantees in the literature assume a balanced mixture. Since imbalanced mixtures are ubiquitous in applications, it is of practical interest to develop MLR methods that are able to handle such settings, as well as corresponding recovery guarantees.

In this paper, we present `Mix-IRLS`, a novel and conceptually different iterative algorithm for MLR, able to handle both balanced and imbalanced mixtures. `Mix-IRLS` is computationally efficient, simple to implement, and scalable to high-dimensional settings. In addition, `Mix-IRLS` successfully recovers the underlying components with relatively few samples, is robust to noise and outliers, and does not require as input the number of components $K$. In Sections 4 and 5 we illustrate the competitive advantage of `Mix-IRLS` over several other methods, on both synthetic and real data.

37th Conference on Neural Information Processing Systems (NeurIPS 2023).

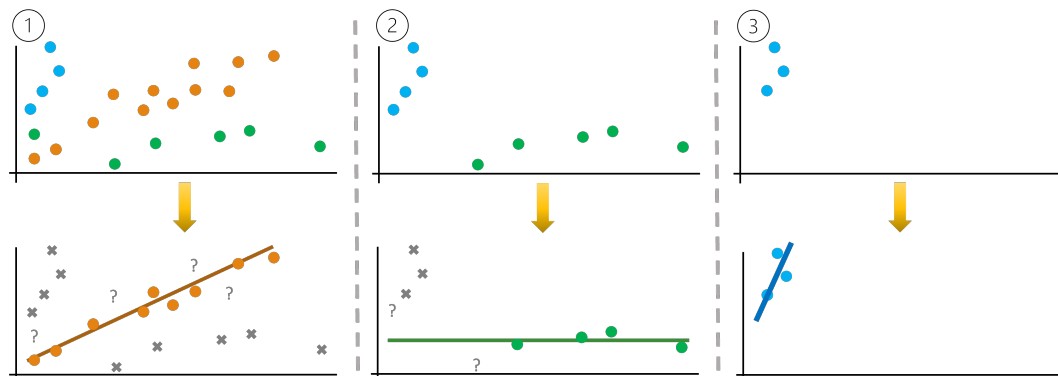

Figure 1: Illustration of `Mix-IRLS`. The data is a mixture of $K = 3$ components. At each round, `Mix-IRLS` excludes samples with poor and moderate fit (marked by 'X' and '?', respectively), and performs linear regression on the good fit samples. The poor fit samples are passed to the next round.

To motivate our approach, let us consider a highly imbalanced mixture, where most samples belong to one model. In this case, the samples that belong to the other models may be regarded as *outliers* with respect to the dominant one. The problem of finding the dominant model may thus be viewed as a specific case of robust linear regression, a problem for which there exist several effective algorithms, see e.g. [27, 50]. After finding the dominant model, we remove its associated samples from the observation set and repeat the process to find the next dominant model. This way, the $K$ linear models are found *sequentially* rather than simultaneously as in the aforementioned methods.

For our `Mix-IRLS` sequential approach to succeed, we found it important to allow in its intermediate rounds an "I don't know" assignment to some of the samples. Specifically, given an estimate of the dominant model, we partition the samples to three classes, according to their fit to the estimated model: good, moderate and poor fit; see Figure 1. The samples with good fit are used to re-estimate the model coefficients; those with poor fit are assumed to belong to a yet undiscovered model, and hence are passed to the next round; the moderate fit samples, on whose model identity we have only low confidence ("I don't know"), are ignored, but used later in a refinement phase.

To recover the dominant model at each of the $K$ rounds, we perform robust regression using iteratively reweighted least squares (IRLS) [24, 10, 40]. IRLS iteratively solves weighted least squares subproblems, where the weight of each sample depends on its residual with respect to the current model estimate. As the iterations proceed, outliers are hopefully assigned smaller and smaller weights, and ultimately ignored; see Figure F.1 for an illustration of this mechanism.

On the theoretical front, in Section 6 we present a recovery guarantee for our method. Specifically, we show that in a population setting with an imbalanced mixture of two components, `Mix-IRLS` successfully recovers the linear models. A key novelty in our result is that it holds for a sufficiently imbalanced mixture rather than a sufficiently balanced one (or even a perfectly balanced one) as is common in the literature [52, 2, 35]. In addition, unlike most available guarantees, our guarantee allows an unknown $K$, and it is insensitive to the initialization, allowing it to be arbitrary.

To the best of our knowledge, our work is the first to specifically handle imbalance in the MLR problem, providing both a practical algorithm as well as a theoretical recovery guarantee. Furthermore, we are the first to propose and analyze an adaptation of IRLS to the MLR problem. A related sequential approach to solve MLR using robust regression was proposed by [3]. They used random sample consensus (RANSAC) instead of IRLS, and without our "I don't know" concept. To find a component, [3] randomly pick $(d + 2)$ samples from the data and run ordinary least squares (OLS) on them, in hope that they all belong to the same component. As discussed by the authors of [3], their approach is feasible only in low dimensional settings, as the probability that all $d + 2$ chosen samples belong to the same component decreases exponentially with the dimension $d$; specifically, [3] considered only $d \leq 5$. In addition, [3] did not provide a theoretical guarantee for their method. In contrast, our `Mix-IRLS` method is scalable to high dimensions, and enjoys a theoretical guarantee.

**Notation.** For a positive integer $K$, denote $[K] = \{1, \ldots, K\}$, and the set of all permutations over $[K]$ by $[K]!$. For a vector $u$, denote its Euclidean norm by $\|u\|$. For a matrix $X$, denote its spectral

---

**Algorithm 1:** `Mix-IRLS`: main phase

**input** : samples $\{(x_i, y_i)\}_{i=1}^n$, number of components $K$, parameters $w_{\text{th}}, \rho, \eta, T_1$
**output :** estimates $\beta_1^{\text{phase-I}}, \ldots, \beta_K^{\text{phase-I}}$

**1** set $S_1 = [n]$
**2 for** $k = 1$ **to** $K$ **do**
**3**     initialize $\beta_k$ randomly
**4**     **repeat** $T_1$ **times**
**5**        compute $r_{i,k} = |x_i^\top \beta_k - y_i|, \quad \forall i \in S_k$
**6**        compute $w_{i,k} = (1 + \eta r_{i,k}^2 / \bar{r}_k^2)^{-1}, \quad \forall i \in S_k$
**7**        compute $\beta_k = (X_{S_k}^\top W_k X_{S_k})^{-1} X_{S_k}^\top W_k \, y_{S_k}$
**8**     **end**
**9**     set $S_{k+1} = \{i \in S_k : w_{i,k} \leq w_{\text{th}}\}$
**10**     set $S_k' = \{\rho \cdot d \text{ samples in } S_k \text{ with largest } w_{i,k}\}$
**11**     **if** $k < K$ *and* $|S_{k+1}| < \rho \cdot d$ **then**
**12**        restart `Mix-IRLS` with $w_{\text{th}} \leftarrow w_{\text{th}} + 0.1$
**13**     **end**
**14**     compute $\beta_k^{\text{phase-I}} = (X_{S_k'}^\top X_{S_k'})^{-1} X_{S_k'}^\top y_{S_k'}$
**15 end**

---

norm by $\|X\|$ and its smallest singular value by $\sigma_{\min}(X)$. Given a matrix $X \in \mathbb{R}^{n \times d}$ and an index set $S \subseteq [n]$, $X_S \in \mathbb{R}^{|S| \times d}$ is the submatrix of $X$ that corresponds to the rows in $S$. Denote by $\text{diag}(w)$ the diagonal matrix $W$ whose entries are $W_{ii} = w_i$. The probability of an event $A$ is $\mathbb{P}[A]$. Denote the expectation and the variance of a random variable $x$ by $\mathbb{E}[x]$ and $\text{Var}[x]$, respectively. The cumulative distribution function of the standard normal distribution $\mathcal{N}(0, 1)$ is $\Phi(\cdot)$.

## 2   Problem Setup

Let $\{(x_i, y_i)\}_{i=1}^n$ be $n$ pairs of explanatory variables $x_i \in \mathbb{R}^d$ and corresponding responses $y_i \in \mathbb{R}$. In standard linear regression, one assumes a linear relation between the response and the explanatory variables, namely $y_i = x_i^\top \beta^* + \epsilon_i$ where $\epsilon_i \in \mathbb{R}$ are random noise terms with zero mean. A common goal is to estimate the coefficient vector $\beta^* \in \mathbb{R}^d$. In mixed linear regression (MLR), in contrast, the assumption is that each response $y_i$ belongs to one of $K$ different linear models $\{\beta_k^*\}_{k=1}^K$. Formally,

$$y_i = x_i^\top \beta_{c_i^*}^* + \epsilon_i, \quad i = 1, \ldots, n, \tag{1}$$

where $c^* = (c_1^*, \ldots, c_n^*)^\top \in [K]^n$ is the label vector. Importantly, we do not know to which component each pair $(x_i, y_i)$ belongs, namely $c^*$ is unknown. For simplicity, we assume the number of components $K$ is known, and later on discuss the case where it is unknown. Given the $n$ samples $\{(x_i, y_i)\}_{i=1}^n$, the goal is to estimate $\beta^* \equiv \{\beta_1^*, \ldots, \beta_K^*\} \subset \mathbb{R}^d$, possibly by estimating also $c^*$. See Figure 4(left) for a real-data visualization of MLR in the simplest setting of $d = 1$ and $K = 2$.

For the $K$ regression vectors $\beta^*$ to be identifiable, sufficiently many samples from each component must be observed. The minimal total number of samples depends on the dimension and the mixture proportions. Denote the vector of mixture proportions by $p = (p_1, \ldots, p_K)$, with $p_k = |\{i \in [n] : c_i^* = k\}|/n$. Then the information limit on the sample size, namely the minimal number of observations $n$ required to recover $\beta^*$ in the absence of noise, is $n_{\text{info}} = d / \min(p)$. As formally proven in Appendix B, no method can recover all $K$ models given fewer samples.

The lack of knowledge of the labels $c^*$ makes MLR significantly more challenging than standard linear regression. Even in the simplified setting of $K = 2$ with perfect balance ($p_1 = p_2 = 1/2$) and no noise ($\epsilon = 0$), the problem is in general NP-hard [52]. However, as detailed in Section 6, various assumptions render the problem computationally tractable, and enable to derive theoretical guarantees.

## 3 The Mix-IRLS Method

For simplicity, we present our algorithm assuming $K$ is known; the case of an unknown $K$ is discussed in Remark 3.3. `Mix-IRLS` consists of two phases. In its first (main) phase, `Mix-IRLS` recursively recovers each of the $K$ components $\beta_1^*, \dots, \beta_K^*$ by treating the remaining components as outliers. The sequential recovery is the core idea that distinguishes `Mix-IRLS` from most other methods. In the second phase, we refine the estimates of the first phase by optimizing them simultaneously, similar to existing methods. As discussed below, in many cases accurate estimates are already found in the first phase, in which case the second phase is not necessary. For brevity, we defer the description of the second phase to Appendix C.

Before we dive into details, let us give a brief overview of the main phase mechanism. At each round of this phase, `Mix-IRLS` uses techniques from robust regression to estimate the largest component present in the data. Then, it partitions the samples to three subsets, according to their fit to the found component: good, moderate and poor. `Mix-IRLS` refines the component estimate using only the samples with good fit, and proceeds to the next round with the poor fit samples. The moderate fit samples are ignored in the main phase, as we have high uncertainty in their component assignment - they either may or may not belong to the found component. As detailed below `Mix-IRLS` depends on four parameters: a model fit threshold $w_{\text{th}} \in (0, 1)$, an oversampling ration $\rho \geq 1$, a tuning parameter $\eta$ and the number of IRLS iterations $T_1 \geq 1$.

Importantly, the robust regression used by `Mix-IRLS` succeeds even if the dominant component consists less than 50% of the data. The well-known "breakdown point" of robust regression at 50% assumes nothing on the data samples, and in particular allows for adversarial outliers. In our case, the 'outliers', namely the samples that belong to a yet undiscovered component, follow an MLR model. As such, standard high-dimensional robust regression methods can recover the underlying linear model even with 70% and 80% outliers; see Appendix H for an empirical illustration.

Next, we present in details the main phase of `Mix-IRLS`. A pseudocode appears in Algorithm 1. Given $X = \begin{pmatrix} x_1 & \cdots & x_n \end{pmatrix}^\top$ and $y = \begin{pmatrix} y_1 & \cdots & y_n \end{pmatrix}^\top$, we initialize the set of active samples to the entire dataset, $S_1 = [n]$. Next, we perform the following procedure for $K$ rounds. At round $k \in [K]$, starting from a random guess $\beta_k$ for the $k$-th vector, we run $T_1$ iterations of IRLS [24]:

$$r_{i,k} = \left| x_i^\top \beta_k - y_i \right|, \, \forall i \in S_k, \qquad \text{(update residuals)} \qquad (2a)$$

$$w_{i,k} = \frac{1}{1 + \eta \cdot r_{i,k}^2 / \bar{r}_k^2}, \, \forall i \in S_k, \qquad \text{(update weights)} \qquad (2b)$$

$$\beta_k = (X_{S_k}^\top W_k X_{S_k})^{-1} X_{S_k}^\top W_k \, y_{S_k}, \quad \text{(update estimate)} \qquad (2c)$$

where $\bar{r}_k = \text{median}\{r_{i,k} \mid i \in S_k\}$, and $W_k = \text{diag}(w_{1,k}, w_{2,k}, \dots)$. The updated estimate (2c) is the minimizer of the weighted least squares objective $\|W^{\frac{1}{2}} (y_{S_k} - X_{S_k} \beta_k)\|^2$. Intuitively, if $\beta_k$ is close to some $\beta_i^*$, then samples from other components receive a lower weight and the updated $\beta_k$ is even closer to $\beta_i^*$. After $T_1$ iterations of (2a)-(2c), we define the subset $S_{k+1}$ of 'poor fit' samples that seem to belong to other components,

$$S_{k+1} = \{i \in S_k : w_{i,k} \leq w_{\text{th}}\}. \qquad (3)$$

This serves as the set of active samples for the next round. In addition, for some $\rho \geq 1$ whose choice is discussed below, we define the subset $S_k'$ of samples with a 'good fit' to the $k$-th component,

$$S_k' = \{\rho \cdot d \text{ samples in } S_k \text{ with largest weights } w_{i,k}\}. \qquad (4)$$

We then refine the estimate of $\beta_k$ by performing OLS on this subset,

$$\beta_k^{\text{phase-I}} = (X_{S_k'}^\top X_{S_k'})^{-1} X_{S_k'}^\top y_{S_k'}. \qquad (5)$$

The subsets $S_k'$ and $S_{k+1}$ are in general disjoint, unless the threshold $w_{\text{th}}$ or the oversampling ratio $\rho$ are too high. The choice for the value of $w_{\text{th}}$ is discussed in Remark 3.1 and Section 6. A suitable value for $\rho$ depends on the ratio between the sample size $n$ and the information limit $n_{\text{info}} = d / \min(p)$. In the challenging setting of $n \approx n_{\text{info}}$, $\rho$ should be set close to 1, confining the algorithm to use the least possible number of samples at each round. If $n \gg n_{\text{info}}$, then $\rho$ can be set to a higher value, allowing the algorithm to estimate each component more accurately and more robustly to noise by using more samples.

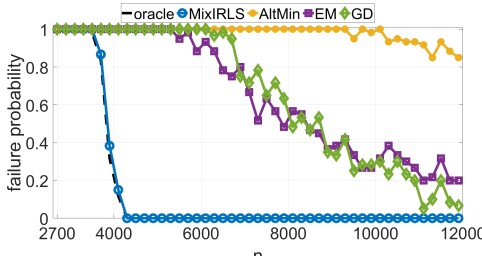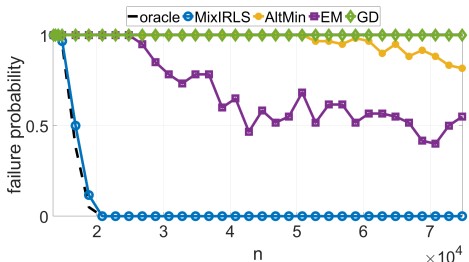

Figure 2: Comparison of various MLR algorithms. Depicted is the failure rate, out of 50 random initializations, for which $F_{\text{latent}} > 2\sigma$ (see (6)), as a function of the sample size $n$. The dimension and noise level are fixed at $d = 300$ and $\sigma = 10^{-2}$. Mixture: $K = 3$ with $p = (0.7, 0.2, 0.1)$ (left panel); $K = 5$ with $p = (0.63, 0.2, 0.1, 0.05, 0.02)$ (right panel).

This concludes the main phase of `Mix-IRLS`. The second (refinement) phase, described in Appendix C, improves the estimates $\beta_k^{\text{phase-I}}$ using also the moderate fit samples that were ignored in the first phase. Yet, empirically in many cases the second phase is not needed, since the main phase already outputs highly accurate estimates of $\beta^*$. This is in accordance with the theoretical result in Section 6.

*Remark* 3.1 (Threshold adaptation). If the threshold $w_{\text{th}}$ is too small, the number of poor fit samples $|S_{k+1}|$ of (3) may be insufficient to estimate the $(k+1)$-th component. To avoid the tuning of $w_{\text{th}}$, we use the following scheme: `Mix-IRLS` gets as input an initial value for $w_{\text{th}}$, and if needed, it increases its value and starts over; see Lines 11 to 13 in Algorithm 1. The value $0.1$ is heuristic, and a more theoretically grounded update rule is an open question for future research.

*Remark* 3.2 (Parameter tuning). `Mix-IRLS` has four input parameters: $\eta, \rho, T_1$ and an initial value for $w_{\text{th}}$ (see Remark 3.1). As empirically demonstrated in Sections 4 and 5, there is no need to carefully tune these parameters; see Appendix F for an explanation of this point. The values used in our experiments are specified in Appendix G.

*Remark* 3.3 (Unknown/overestimated $K$). If $K$ is unknown, or only an upper bound $K_{\max}$ is given, `Mix-IRLS` can automatically find the true $K$ without cross validation. To find it, we ignore the resetting criterion (Lines 11 to 13 in Algorithm 1), and instead run `Mix-IRLS` until there are too few samples to estimate the next component, namely $|S_{k+1}| < \rho d$. We then set $K$ to the number of components with at least $\rho d$ associated samples, and proceed to the second phase.

## 4 Simulation Results

We present simulation results on synthetic data in this section, and on several real-world datasets in the next one. We compare the performance of `Mix-IRLS` to the following algorithms: (i) `AltMin` - alternating minimization [52, 53]; (ii) `EM` - expectation maximization [5, Chapter 14], [19]; and (iii) `GD` - gradient descent on a factorized objective [56]. We implemented all methods in MATLAB.[1] In some of the simulations, we additionally ran a version of `EM` for which the mixture proportions $p$ are given as prior knowledge, but it hardly improved its performance and we did not include it in our results. In addition, we plot the performance of an oracle which is provided with the vector $c^*$ of the true labels of all $n$ samples and separately computes the OLS solution for each component.

All methods were given the same random initialization, as described shortly. Other initializations did not qualitatively change the results. For `EM` and `GD` we added an `AltMin` refinement step at the end of each algorithm to improve their estimates. In principle, `Mix-IRLS` has several tuning parameters. However, `Mix-IRLS` performs well with a fixed set of values (see Remark 3.2). Hence, in the following, we show the results for a tuning-free variant of our method. Further technical details, including maximal number of iterations, early stopping criteria and parameter tuning of `GD` appear in Appendix G.

Similar to [56, 21], in each simulation we sampled the entries of the explanatory variables $X$ and of the regression vectors $\beta^*$ from the standard normal distribution $\mathcal{N}(0, 1)$. In this section, the dimension is fixed at $d = 300$. The additive noise terms are Gaussian, $\epsilon_i \sim \mathcal{N}(0, \sigma^2)$. Additional simulations with other values of $d$ and $\sigma$, including different noise levels for the $K$ components (i.e.,

---

[1]MATLAB and Python code implementations of `Mix-IRLS` are available at `github.com/pizilber/MLR`.

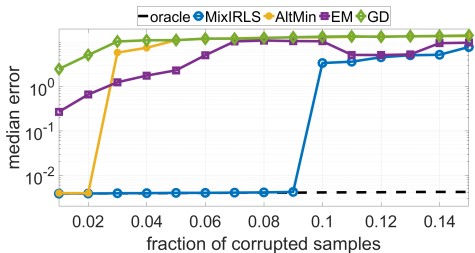
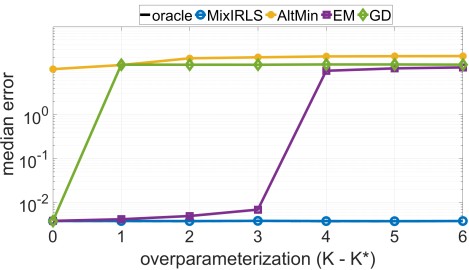

Figure 3: Robustness of several MLR algorithms to outliers (left panel) and to overparameterization (right panel). Values of $d, \sigma, K$ and $p$ are the same as in Figure 2(left), and $n = 12000$. Qualitatively similar results for a balanced mixture appear in Figures H.15 and H.16. X-axis on left panel is fraction of outliers; X-axis on right panel is the difference between the number of components $K$ given to the algorithms and the true $K^*$.

different $\sigma_1, \dots, \sigma_K$), as well as simulations with different separation levels between the components, appear in Appendix H. As discussed in the introduction, a central motivation for the development of `Mix-IRLS` is dealing with imbalanced mixtures. Hence, in this section, the labels $c_i^*$ were drawn from a multinomial distribution with highly imbalanced proportions. In Appendix H, we present results for balanced and moderately imbalanced mixtures.

We measure the accuracy of an estimate $\beta \equiv \{\beta_1, \dots, \beta_K\}$ by the following quantity:

$$F_{\text{latent}}(\beta; \beta^*) = \min_{\sigma \in [K]!} \max_{k=1,\dots,K} \|\beta_{\sigma(k)} - \beta_k^*\|. \tag{6}$$

The minimization above is over all $K!$ permutations, which makes the accuracy measure invariant to the order of the regression vectors in $\beta$. A similar objective was used by [53, 56].

All algorithms were initialized with the same random vectors $\beta_1, \dots, \beta_K$, whose entries were sampled from the standard normal distribution $\mathcal{N}(0, 1)$. For each simulation, we performed 50 independent realizations, each with a different random initialization $\beta$, and report the median errors and the failure probability. The latter quantity is defined as the percentage of runs whose error $F_{\text{latent}}$ (6) is above $2\sigma$; see an explanation for this choice in Appendix G. Due to space limits, some figures appear in Appendix H.

In the first simulation, we examine the performance of the algorithms as a function of the sample size $n$. The results are depicted in Figure 2, and the corresponding runtimes in Figure H.3. `Mix-IRLS` is shown to recover the components with sample size very close to the oracle's minimum. All competing methods, in contrast, get stuck in bad local minima unless the sample size is much larger. Importantly, this behavior does not follow from the presence of noise, and as shown in Appendix H, the result does not qualitatively change in a noiseless setting. Moreover, as shown in Figure H.8, the nearly optimal performance of `Mix-IRLS` seems to be independent of the dimension $d$. It does depend, however, on the mixture proportions: for a moderately imbalanced mixture, `Mix-IRLS` does not match the oracle performance. Yet, even in this case, `Mix-IRLS` still markedly outperforms the other methods; see Figure H.6.

Next, we explore the robustness of the algorithms to additive noise, outliers and overestimation. Figure H.10 shows that all algorithms are stable to additive noise of various levels, but only `Mix-IRLS` matches the oracle performance in all runs. To study robustness to outliers, in the following simulation we artificially corrupt a fraction $f \in (0, 1)$ of the observations. A corrupted response $\tilde{y}_i$ is sampled from a normal distribution with zero mean and variance $\sum_{j=1}^n y_j^2 / n$, independently of the original value $y_i$. Figure 3(left) shows the error (6) of the algorithms as a function of the corruption fraction $f$. To let the algorithms cope with outliers while keeping the comparison fair, we made the *same* modification in all of them: at each iteration, the estimate $\beta$ is calculated based on the $\lceil (1 - f)n \rceil$ samples with smallest residuals. In `Mix-IRLS`, we implemented this modification only in the second phase. As shown in the figure, empirically, `Mix-IRLS` can deal with a corruption fraction of $f = 9\%$, which is over 4 times more corrupted samples than the other algorithms. In the balanced setting, `Mix-IRLS` can deal with roughly twice as many corrupted samples ($f = 17\%$), which is almost 6 times more outliers than other methods. These results should not be surprising in light of the fact that robust regression is at the heart of `Mix-IRLS`'s mechanism.

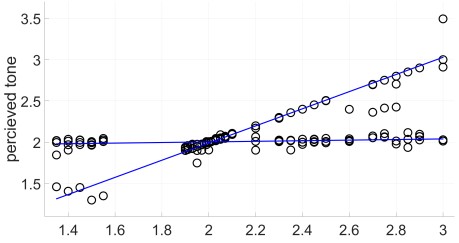
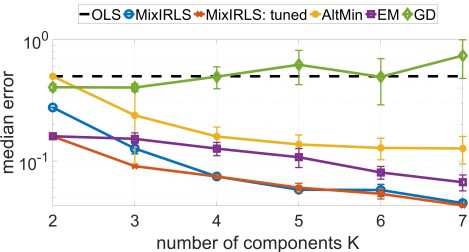

Figure 4: Left panel: `Mix-IRLS` estimate for the music perception data [13], where $d = 1$ and $n = 150$. Right panel: Comparison of several MLR algorithms on the medical insurance dataset. Note that the y-axis is on a log scale. Median estimation errors are calculated across 50 random initializations according to (8), and error bars correspond to the median absolute deviation.

The last simulation considered the case of an unknown number of components. Specifically, the various algorithms were given as input a number of components $K \geq K^*$ where $K^*$ is the true number of mixture components in the data. Here, the error is defined similar to (6), but with $K^*$ instead of $K$. Namely, the error is calculated based on the best $K^*$ vectors in $\beta$, ignoring its other $K - K^*$ vectors. Figure 3(right) shows that most algorithms have similar performance at the correct parameter value $K = K^*$, and `EM` succeeds also at small overparameterization, $K - K^* \leq 3$. Only `Mix-IRLS` is insensitive to overparameterization, and succeeds with unbounded $K$. This feature is attained thanks to the sequential nature of `Mix-IRLS` (Remark 3.3). Similar results hold for balanced mixtures; see Appendix H.

## 5   Real-World Datasets

We begin by analyzing the classical music perception dataset of Cohen [13]. In her thesis, Cohen investigated the human perception of tones by using newly available electronic equipment. The $n = 150$ data points acquired in her experiment are shown in Figure 4(left). Cohen discussed two music perception theories: One theory predicted that in this experimental setting, the perceived tone (y-axis) would be fixed at $2.0$, while the other theory predicted an identity function ($y = x$). The results, depicted in Figure 4(left), support both theories. As a mathematical formulation of this finding, Cohen proposed the MLR model (1) with $K = 2$, where the labels $c^*$ are i.i.d. according to a Bernoulli distribution; see also [15].

In Figure 4(left), the untuned version of `Mix-IRLS` is shown to capture the two linear trends in the data. Notably, untuned `Mix-IRLS` was not given the number of components, but automatically stopped at $K = 2$ with its default parameters. By increasing the sensitivity of `Mix-IRLS` to new components via the parameter $w_{\text{th}}$, it is possible to find three or even more components; see Figure I.19.

Next, we compare the performance of `Mix-IRLS` to the algorithms listed in the previous section on a more challenging problem: the CO2 emission by vehicles in Canada dataset, available on Kaggle (see Appendix G). This dataset approximately follows an MLR model, where the samples can be clustered into components according to their fuel type: regular gasoline, premium gasoline, diesel or ethanol ($K = 4$). The mixture is imbalanced, with the proportions $p = (0.49, 0.43, 0.05, 0.03)$. We ask the following question: given the engine features (engine size, number of cylinders, fuel consumption in city and highway), how close are the methods' clusters to the (ground-truth) fuel type clusters?

We measure the performance of the methods with class-wise balanced accuracy, defined as the average of sensitivity/recall and specificity. We chose this measure as it is considered suitable for imbalanced mixtures, but other measures yield qualitatively similar results. As clustering is defined only up to permutation, for each method we choose the permutation that maximizes the overall intersection of its estimated clusters with the true ones, defined as

$$\max_{\sigma \in [K]!} \frac{1}{n} |\{i \in [n] : \sigma(c_i) = c_i^*\}| . \tag{7}$$

The results, reported in Table 1, demonstrate the effectiveness of `Mix-IRLS` on imbalanced mixtures. While on the dominant class (regular gasoline) all methods achieve comparable balanced accuracy,

Table 1: Results of MLR algorithms on the CO2 emission from vehicles dataset. Reported are median balanced accuracy and median absolute deviation based on 50 random initializations.

| Balanced accuracy | Mix-IRLS | AltMin | EM | GD |
|---|---|---|---|---|
| regular gasoline (49%) | $\mathbf{0.58 \pm 0.00}$ | $\mathbf{0.59 \pm 0.03}$ | $\mathbf{0.56 \pm 0.04}$ | $\mathbf{0.57 \pm 0.07}$ |
| premium gasoline (43%) | $\mathbf{0.59 \pm 0.00}$ | $0.48 \pm 0.10$ | $0.48 \pm 0.01$ | $0.40 \pm 0.03$ |
| diesel (5%) | $\mathbf{0.89 \pm 0.00}$ | $\mathbf{0.78 \pm 0.13}$ | $0.54 \pm 0.07$ | $0.70 \pm 0.12$ |
| ethanol (3%) | $\mathbf{0.74 \pm 0.00}$ | $0.63 \pm 0.07$ | $\mathbf{0.77 \pm 0.08}$ | $0.61 \pm 0.05$ |

`Mix-IRLS` significantly outperforms the other methods on the remaining classes. `EM` is the only method that is slightly better than `Mix-IRLS` on one of the rare classes, but achieves markedly worse balanced accuracy on the other two rare classes. Moreover, importantly, `Mix-IRLS` is the only method that is stable with respect to the initialization, demonstrating negligible variance across different random initializations.

Finally, we compare the performance of the algorithms on four of the most popular benchmark datasets for linear regression, all of which are available on Kaggle (see Appendix G): medical insurance cost, red wine quality, World Health Organization (WHO) life expectancy, and fish market. The task in each dataset is to predict, respectively: medical insurance cost from demographic details; wine quality from its psychochemical properties; life expectancy from demographic and medical details; and fish weight from its dimensions. For these datasets, MLR is at best an approximate model, and its regression vectors $\beta^*$ are unknown. Hence, we replace (6) by the following quality measure which represents the fit of an MLR model to the data:

$$F_{\text{real}}(\beta; X, y) = \frac{1}{\text{Var}[y]} \cdot \frac{1}{n} \sum_{i=1}^{n} \min_{j \in [K]} (x_i^\top \beta_j - y_i)^2, \tag{8}$$

resembling the K-means objective for clustering [23]. To illustrate the advantage of a multi-component model, we also report the results of an ordinary least squares (OLS) solution. In this section, besides vanilla `Mix-IRLS` we report the results of a tuned variant, denoted `Mix-IRLS:tuned`. The tuning is done using only the observed objective (8), and no oracle is involved in the procedure.

As the number of components $K$ is unknown, we explore the algorithms' performance given different values of $K$, ranging from 2 to 7. The upper limit 7 was chosen arbitrarily, and the results do not change qualitatively for larger values. All algorithms start from the same random initialization. Additional details appear in Appendix G.

Figure 4(right) shows the performance of the algorithms on the medical insurance dataset in terms of the median error (8) across 50 different random initializations. The results for the other three datasets, as well as the minimal errors achieved across the realizations, are deferred to Appendix I. In general, both `Mix-IRLS` and `Mix-IRLS:tuned` improve upon the other methods, sometimes by 30% or more.

The experiments on the Kaggle datasets demonstrate that empirically `Mix-IRLS` finds MLR coefficients that fit the data significantly better than other methods. Hence, given as input data that (approximately) does follow an MLR model for some $K$, our method would often compute accurate estimates of its corresponding mixture coefficients where other methods would fail to do so.

## 6 Recovery Guarantee for Mix-IRLS

In this section, we theoretically analyze `Mix-IRLS` in a population setting with an infinite number of samples. For simplicity, we assume the explanatory variables are normally distributed. Without loss of generality, we assume zero mean and identity covariance,

$$x_i \sim \mathcal{N}(0, I_d). \tag{9}$$

In the more general case where $x_i \sim \mathcal{N}(\mu, \Sigma)$ for some $\mu$ and $\Sigma$, one may find $\mu$ and $\Sigma$ from the data, and mean center and whiten the samples $x_i \to \Sigma^{-1/2}(x_i - \mu)$, before applying our method. The responses $y_i$ are assumed to follow model (1) with $K = 2$ components $(\beta_1^*, \beta_2^*)$ and labels $c_i^*$

generated independently of $x_i$ according to mixture proportions $p_1 \geq p_2 > 0$. This setting was considered in several previous theoretical works that analyzed the EM method for MLR [2, 14, 33, 36].

We assume the noise terms $\epsilon_i$ are all i.i.d., zero-mean random variables, independent of $x_i$ and $c_i^*$. We further assume they are bounded and follow a symmetric distribution,

$$|\epsilon_i| \leq \sigma_\epsilon \quad \text{and} \quad \mathbb{P}[\epsilon_i \leq t] = \mathbb{P}[\epsilon_i \geq -t] \quad \forall t. \tag{10}$$

For analysis purposes, we consider a slightly modified variant of Mix-IRLS, formally described in Appendix E. In this variant, Mix-IRLS excludes samples $x_i$ with large magnitude, $\|x_i\|^2 > R$ where $R$ is a fixed parameter. A natural choice for its value is $R \sim \mathbb{E}[\|x_i\|^2] = d$, e.g. $R = 2d$. In a high-dimensional setting with $d \gg 1$, such a choice excludes a proportion of the samples that is exponentially small in $d$. For simplicity, we present our result assuming $R$ is large, corresponding to large dimension $d$; the general result appears in Lemma E.1.

The following theorem states that given a sufficiently imbalanced mixture, namely $p_1$ is large enough, Mix-IRLS successfully recovers the underlying vectors $\beta_1^*$ and $\beta_2^*$.

**Theorem 6.1.** *Let* $\{(x_i, y_i)\}_{i=1}^\infty$ *be i.i.d. from a mixture of* $K = 2$ *components with proportions* $(p_1, p_2)$*, regression vectors* $(\beta_1^*, \beta_2^*)$*, and noise terms* $\epsilon_i$ *that follow* (10) *with* $\sigma_\epsilon$*. Denote* $\Delta = \beta_1^* - \beta_2^*$*, and* $\gamma = 5p_2/4$*. Suppose that*

$$q \equiv \gamma + \left(1 + \frac{1}{\sqrt{R}}\right) \frac{\sigma_\epsilon}{\|\Delta\|} < \frac{1}{2}. \tag{11}$$

*Further assume that* Mix-IRLS *is run with parameters* $\rho = \infty$*, and with* $\eta$*,* $w_{th}$ *that satisfy,*

$$\frac{1}{1 + \eta(1-q)^2\|\Delta\|^2} < w_{th} < \frac{1}{1 + \eta q^2\|\Delta\|^2}, \tag{12}$$

*Then for sufficiently large* $R$*, starting from an arbitrary initialization, the first phase of* Mix-IRLS *with at least one iteration* ($T_1 \geq 1$*) recovers* $\beta^*$ *up to an error that decreases with increasing* $R$*,*

$$\max_{k=1,2} \|\beta_k - \beta_k^*\| \leq \frac{1}{\sqrt{R}} \frac{\sigma_\epsilon}{\sigma_\epsilon + \gamma\|\Delta\|}. \tag{13}$$

*Specifically, in the absence of noise* ($\sigma_\epsilon = 0$*),* Mix-IRLS *perfectly recovers the regression vectors.*

Theorem 6.1 considers only the first phase of Mix-IRLS, as it is sufficient to recover the regression vectors in the described setting. Indeed, empirically, the second phase is often unnecessary. The choice of an oversampling ratio $\rho = \infty$ is suited to the population setting where $n = \infty$; see the discussion after Eq. (5). The theorem proof appears in Appendix E.

*Remark* 6.2 (Different noise levels). Theorem 6.1 holds also in the case of different noise levels $\sigma_1, \sigma_2$ for each of the two mixture components. In this case, $\sigma_\epsilon$ is replaced by $\max\{\sigma_1, \sigma_2\}$. Indeed, as illustrated empirically in in Figure H.11, Mix-IRLS is able to handle such a case as well.

*Remark* 6.3 (Required imbalance). Due to (11), Theorem 6.1 holds only for a sufficiently imbalanced mixture. In the absence of noise, the factor $5/4$ in the definition of $\gamma$ may be relaxed to any number arbitrarily close to one, at the expense of increasing $R$. In turn, this implies that our theorem holds for any mixture with the probability of the second component allowed to be arbitrarily close to $p_2 < 1/2$. In simple words, in the noiseless setting our guarantee holds as long as the mixture is not perfectly balanced. More generally, in the presence of noise, there is a trade-off between mixture imbalance and noise level: the proportion of the dominant component $p_1$ increases with the noise level $\sigma_\epsilon$. This theoretical finding is in agreement with the simulation results in Section 4: Mix-IRLS works better as the imbalance increases. We emphasize that empirically, Mix-IRLS works well also on balanced mixtures, $p_1 = p_2 = 1/2$; see Appendix H. Hence it is an open problem to derive theoretical guarantees for a perfectly balanced mixture.

*Remark* 6.4 (Allowed range for $w_{th}$). Theorem 6.1 holds for a limited range of values for $w_{th}$, see (12). This range depends on $q$, which in turn depends on the noise level and the mixture imbalance. For example, at a noise level $\sigma_\epsilon = 10^{-2}$, proportions $p = (3/4, 1/4)$, separation of $\|\Delta\| = 1$, parameter choice of $\eta = 1$ and $R \geq 2$, the range for which recovery is guaranteed is $0.69 \leq w_{th} \leq 0.9$. As mentioned previously, empirically we run Mix-IRLS with an initial value of $w_{th} = 0.1$ which is increased by the algorithm (see lines 11-13 in Algorithm 1). Indeed, in accordance to this analysis, in many cases the final value is within this range.

*Remark* 6.5 (Overparameterization / unknown $K$). In practical scenarios, the number of components $K$ is often unknown. Remarkably, Theorem 6.1 can be extended to an overparameterized setting, where the true number of components $K^*$ is 2 but `Mix-IRLS` is given an overestimated number $K > 2$, together with a corresponding (arbitrary) initialization $(\beta_1, \ldots, \beta_K)$. This is explicitly discussed in Appendix D (Proposition D.1), and also demonstrated empirically in Figure 3(right).

Theorem 6.1 and Remark 6.5 are in accordance with several empirical findings from previous sections: `Mix-IRLS` performs better on imbalanced mixtures than on balanced ones; it copes well with an overparameterized $K$; and it works well starting from a random initialization. Our analysis (Appendix E) sheds light on the inner mechanism of `Mix-IRLS` that enables these features.

**Comparison to prior work.** Several works derived MLR recovery guarantees for `AltMin` [52, 53, 21] and `GD` [56, 38] in a noiseless setting. More related to our Theorem 6.1 are works that studied the population `EM` algorithm in the presence of noise [2, 14, 33, 36, 34]. These works assumed a perfectly balanced mixture of two components, $p_1 = p_2 = 1/2$. An exception is [34], who allowed for $K > 2$ and an imbalanced mixture. However, their allowed imbalance is limited. In addition, they required a sufficiently accurate initialization. A key novelty in our result is not only that we allow for highly imbalanced mixtures, but that large imbalance actually makes recovery *easier* for `Mix-IRLS`: since the quantity $q$ of (11) is monotonically decreasing with the mixture imbalance, the allowed range (12) of the parameter $w_{\text{th}}$ increases with the imbalance. Furthermore, our result holds for an arbitrary initialization. The downside is that Theorem 6.1 requires sufficient imbalance (see Remark 6.3), and does not provide a recovery guarantee for our method on a perfectly balanced mixture, even though empirically, our method works well also in this case. Our result is novel in another aspect as well. In contrast to most existing guarantees, Theorem 6.1 holds also for an arbitrary input number of components $K$; see Remark 6.5.

`Mix-IRLS` recovers the first component by treating the samples from the second component as outliers. In the noiseless setting, our guarantee allows the second component to consist almost $1/2$ of the data; see Remark 6.3. For comparison, in the context of robust regression, [40] recently analyzed an IRLS method in noiseless setting, and allowed less than $1/5$ corrupted samples. Our higher tolerance is possible thanks to the strong structural assumption of MLR (1).

## 7 Summary and Discussion

In this work, we presented a novel method to solve MLR, `Mix-IRLS`, that handles both imbalanced and balanced mixtures. `Mix-IRLS` is robust to outliers and to an overestimated number of components. The latter feature leads to another contribution of our method: under suitable conditions, it can be run with an overestimated $K > K^*$ and will automatically find the true number of components $K^*$.

The basic idea of `Mix-IRLS` - sequential recovery using tools from robust regression - was also employed by [3]. Several important differences between their method and `Mix-IRLS` were listed in the introduction; first and foremost is the fact that [3] is limited to low dimensions. It is interesting to note that [1] also made a connection between MLR and robust regression, but the other way around: as a simplified theoretical model, they assumed the outliers follow a linear model, and applied the `EM` algorithm to the obtained MLR problem to detect them. Also relevant to our work are recent papers on subspace clustering [48, 18], list-decodable learning [9, 31] and real phase retrieval [42]. We discuss their relation to our work, additional relevant papers and future research directions, in Appendix A.

Our current theoretical analysis is limited to the case of only $K = 2$ components and a population setting with infinitely many samples. While the first assumption is very common in the literature (e.g., [2, 36], and many others), population analysis is usually followed by a finite-sample one. In our case, a main challenge in such an analysis revolves around the sequential nature of `Mix-IRLS`. Once `Mix-IRLS` estimates a component, it removes its associated samples. Analyzing a modified sample set introduces complex statistical dependencies - the data no longer follow a normal distribution but rather a conditional distribution. In the population setting, various quantities happen to cancel out nicely; see, for example, the derivation of (E.17) from (E.16). Under the setting of finite sample size, we need to bound these quantities under a conditional distribution. We hope to be able to overcome these challenges and extend our analysis to a finite-sample setting in the future. Finally, another interesting future research direction is to formally prove `Mix-IRLS`'s robustness to outliers, as was empirically demonstrated in Section 4.

## Acknowledgements

The research of P.Z. was partially supported by a fellowship for data science from the Israeli Council for Higher Education (CHE). B.N. is the incumbent of the William Petschek Professorial Chair of Mathematics. The research of B.N. was supported in part by grant 2362/22 from the Israel Science Foundation. We thank Yuval Kluger and Ofir Lindenbaum for interesting discussions. We thank the authors of [56] for sharing their code with us. We also thank the anonymous reviewers for their valuable feedback that improved the quality of our manuscript.

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
