# Supplementary Material:
# Imbalanced Mixed Linear Regression

## A   Additional Related Work and Future Research Directions

The most popular approach to solve MLR is, arguably, expectation-maximization and its variants. In recent years, this approach was extensively studied both theoretically and empirically. Most works on EM [19, 26, 2, 14, 33, 36, 34, 55, 35] made two simplifying assumptions: (i) Gaussian noise, $\epsilon \sim \mathcal{N}(0, \sigma^2 I)$; and (ii) model-free clustering. The second assumption means that the cluster assignment $c_i^*$ is random and independent of the sample location in space $x_i$. We made similar assumptions in our work. Several other works on expectation-maximization extended this setting in both directions: (i) non-Gaussian noise [32, 28, 46, 51, 25, 4], and (ii) model-based clustering, where $c_i^*$ potentially depends on $x_i$ [54, 28, 29]. Adapting `Mix-IRLS` to these settings is an interesting research direction.

Besides expectation-maximization, other approaches proposed in the literature include alternating minimization [52, 53, 21, 41], convex relaxation [12, 22, 30], and gradient descent applied to a suitable objective [56, 38, 17]. These methods, as well as expectation-maximization, recover the linear models simultaneously, and the corresponding works did not pay specific attention to the imbalanced MLR setting. In particular, all the available theoretical guarantees in the literature either assume a perfectly balanced mixture [52, 12, 2, 14, 33, 36, 21, 17, 35], or at least a sufficiently balanced one [53, 56, 38, 11, 34]. In sharp contrast, our guarantee holds for a sufficiently *imbalanced* mixture.

The MLR problem is related to several other learning problems. One special case of MLR, known as real phase retrieval, is where the number of components is two ($K = 2$) and the regression vectors are equal up to sign ($\beta_1^* = -\beta_2^*$). Recently, [42] described an application of IRLS with an $\ell_p$ objective to this problem. As finding a single vector (either $\beta_1^*$ or $\beta_2^*$) is sufficient to solve real phase retrieval, their IRLS is not followed by a sequential recovery of other components as in our approach.

MLR is closely related to list-decodable learning, introduced by [9]. In this framework, out of $n$ observed data points, only $\alpha n$ are drawn from a distribution of interest ($\alpha < 1$), and the rest are arbitrary. Obviously, in case of $\alpha \le 1/2$, it is impossible to uniquely find the true distribution. Instead, the goal is to find a list of length $\text{poly}(1/\alpha)$ of possible solutions. Recently, [31] proposed and analyzed an algorithm to solve this problem in the special case of linear regression, where the $\alpha n$ samples follow a linear relation $y_i = x_i^\top \beta + \epsilon_i$. It would be interesting to generalize our `Mix-IRLS` method to deal with this problem as well.

Under certain assumptions (e.g. model-free clustering), MLR can be viewed as a special case of subspace clustering; see [56, 43]. The setting of one-dimensional subspaces, known as hyperplane clustering, is even more related to the clustering task in MLR. Similar in spirit to out work, [48, 18] proposed a sequential approach to solve hyperplane clustering. Specifically, [48] employed the IRLS method, albeit with an objective of $\ell_p$ minimization; our weighting scheme is different, and in particular involves a tuning parameter $\eta$. Other methodological differences between [48] and our work are the use of "I don't know" assignments and the simultaneous phase of `Mix-IRLS`. Finally, [48] did not theoretically analyze their IRLS method.

Submitted to 37th Conference on Neural Information Processing Systems (NeurIPS 2023). Do not distribute.

More generally, MLR is a special case of finite mixture models; see [39] for a comprehensive review on this broader field. The framework presented in this paper can, in principle, be applied to other, non-linear mixture models: given a robust non-linear regressor, we can use it to separate the components of the mixture. This is another appealing direction for future research.

# B  The information limit

**Proposition B.1.** *Given $n < n_{info} \equiv d/\min(p)$ samples, the MLR problem is not identifiable.*

*Proof.* In linear regression with $d$-dimensional explanatory variables, one must observe at least $d$ samples, otherwise the system is underdetermined. Hence, in MLR, one must observe at least $d$ samples from each component. Since the number of samples associated with the $k$-th component is $n \cdot p_k$, one must have $n \cdot p_k \geq d$, $\forall k$, from which the claim follows. □

# C  The Second Phase of `Mix-IRLS`

The output of the first phase of `Mix-IRLS` are estimates $\beta_1^{\text{phase-I}}, \dots, \beta_K^{\text{phase-I}}$ for the regression vectors. In the second phase, we initialize $\beta = \beta^{\text{phase-I}}$, and then run the following scheme for $T_2$ iterations. A pseudocode appears in Algorithm C.1. First, we calculate the following residuals and modified weights,

$$r_{i,k} = \left| x_i^\top \beta_k - y_i \right|, \quad \forall i \in [n], \forall k \in [K], \tag{C.1a}$$

$$\tilde{w}_{i,k} = \frac{1/(r_{i,k}^2 + \epsilon_{\text{mp}})}{\sum_{k'=1}^K 1/(r_{i,k'}^2 + \epsilon_{\text{mp}})}, \quad \forall i \in [n], \forall k \in [K], \tag{C.1b}$$

where $\epsilon_{\text{mp}}$ is the machine precision. Next, we binarize some of the weights in a two-step scheme. Let

$$H = \left\{ i \in [n] : \exists k \in [K] \text{ s.t. } \tilde{w}_{i,k} \geq \frac{2}{3} \right\} \tag{C.2}$$

be the subset of samples with a single dominant weight. The value $2/3$ is arbitrary, and the performance of `Mix-IRLS` is insensitive to its exact value. (i) For each sample in $H$, we set its highest weight to $1$ and zero out the others; (ii) for the samples outside $H$, we zero out the weights smaller than $1/K$, and renormalize $\tilde{w}_{i,k} = \tilde{w}_{i,k} / \sum_{k'=1}^K \tilde{w}_{i,k'}$. Finally, we calculate a weighted least squares,

$$\beta_k = (X^\top \tilde{W}_k X)^{-1} X^\top \tilde{W}_k y, \quad \forall k \in [K], \tag{C.3}$$

where $\tilde{W}_k = \text{diag}(\tilde{w}_{1,k}, \tilde{w}_{2,k}, \dots)$. We iterate Equations (C.1) to (C.3) $T_2$ times. This concludes the second phase of `Mix-IRLS`. Note that for $K = 2$, the second phase coincides with the alternating minimization algorithm [52]. The final output of `Mix-IRLS` is $\beta = (\beta_1, \dots, \beta_K)$.

Both phases of `Mix-IRLS` employ an IRLS approach. However, as discussed earlier, they are fundamentally different: the first phase estimates the components sequentially, while the second one does it simultaneously. In particular, this leads to a slightly modified weighting scheme. Equation (2b) of the first phase of `Mix-IRLS` employs a standard Cauchy weighting, which was found empirically to suit our needs; other standard schemes could be used as well. The scheme is similar in the second phase but involves two differences tailored to our needs. First, since the weights are calculated for all components, they are subsequently normalized over the components such that $\sum_{k=1}^K w_{i,k} = 1$. Due to this normalization, we do not need the weights to lie between $0$ and $1$ as in standard weighting schemes; so instead of $1/(r^2 + 1)$ (Cauchy) we use $1/(r^2 + \epsilon_{\text{mp}})$ (C.1b). This modification encourages the dominance of one component over the others.

*Remark* C.1 (Computational complexity). Each round of the first phase of `Mix-IRLS` is dominated by the weighted least squares problem (2c), whose complexity is $\mathcal{O}(n^2 d)$. The complexity of the first phase is thus $\mathcal{O}(n^2 dK T_1)$. Similarly, as each round in the second phase is dominated by the weighted least squares computation (C.3), its complexity is $\mathcal{O}(n^2 dK T_2)$. The overall number of operations in `Mix-IRLS` is thus $\mathcal{O}\left(n^2 dK(T_1 + T_2)\right)$.

---

**Algorithm C.1:** `Mix-IRLS`: refinement phase (often unnecessary)

---

**input** : samples $\{(x_i, y_i)\}_{i=1}^n$, number of components $K$, number of iterations $T_2$, phase I estimates $\{\beta_k^{\text{phase-I}}\}_{k=1}^K$

**output :** estimates $\beta_1, \ldots, \beta_K$ such that $y_i \approx x_i^\top \beta_{k(i)}$ for some function $k : [n] \to [K]$

---

1 initialize $\beta_k = \beta_k^{\text{phase-I}}, \quad \forall k \in [K]$

2 **for** $t = 1$ **to** $T_2$ **do**

3     compute $r_{i,k} = |x_i^\top \beta_k - y_i|, \quad \forall i \in [n], \forall k \in [K]$

4     compute $\tilde{w}_{i,k} = (r_{i,k}^2 + \epsilon_{\text{mp}})^{-1} / \sum_{k'=1}^K (r_{i,k'}^2 + \epsilon_{\text{mp}})^{-1}, \quad \forall i \in [n], \forall k \in [K]$

5     set $H = \{i \in [n] : \exists k \in [K] \text{ s.t. } \tilde{w}_{i,k} \geq 2/3\}$

6     set $\tilde{w}_{i,k} = 1$ if $\tilde{w}_{i,k} = \max_{k'} \tilde{w}_{i,k'}$ and 0 otherwise, $\quad \forall i \in H, k \in [K]$

7     set $\tilde{w}_{i,k} = 0, \quad \forall i, k \text{ s.t. } \tilde{w}_{i,k} < 1/K$

8     compute $\tilde{w}_{i,k} = \tilde{w}_{i,k} / \sum_{k'=1}^K \tilde{w}_{i,k'}, \quad \forall i \in [n], \forall k \in [K]$

9     compute $\beta_k = (X^\top \tilde{W}_k X)^{-1} X^\top \tilde{W}_k y, \quad \forall k \in [K]$

10 **end**

---

## D    Theoretical Guarantee with an Unknown $K$

In Remark 3.3 of the main text, we claimed that Theorem 6.1 can be extended to the unknown $K$ setting, where the true $K$ is 2 but `Mix-IRLS` is unaware of it. Proposition D.1 formulates this claim.

**Proposition D.1.** *Assume the conditions of Theorem 6.1, but with $K = 2$ not given as input to* `Mix-IRLS`*. Then* `Mix-IRLS` *would correctly stop the IRLS scheme* (2) *after two rounds according to the stopping criterion described in Remark 3.3. Moreover, the resulting estimator has the same error in this case as in the known $K$ case.*

Intuitively, this happens as the second round recovers the second regression vector quite accurately, so that removing the samples with good and moderate fit actually removes all the samples and leaves no active samples for a third round. The formal proof appears in Appendix E.1.

## E    Proof of Theorem 6.1

Let us first describe the modified algorithm which we analyze theoretically. For simplicity, we suit it to the assumptions of Theorem 6.1, namely $K = 2$ components and $T_1 = 1$ iterations. For clarity, a full pseudocode appears in Algorithm E.2.

First, we replace the original formula of the weights in (2b). Instead of scaling the residuals by the square median residual $\bar{r}_k^2$, we assume the following formula:

$$w_{i,k} = \frac{1}{1 + \eta r_{i,k}^2 / R}, \tag{E.4}$$

where $R \geq 1$ is a constant. In addition, we change the definition of the subsets in (3)-(4) as follows:

$$S_2 = S_2' = \{i \in S_1 : \|x_i\|^2 \leq R \text{ and } w_{i,1} \leq w_{\text{th}}\}, \tag{E.5a}$$

$$S_1' = \{i \in S_1 : \|x_i\|^2 \leq R \text{ and } w_{i,2} \leq w_{\text{th}}\}. \tag{E.5b}$$

The equality $S_2 = S_2'$ corresponds to taking $\rho = \infty$ in (4). As discussed after (4), the oversampling ratio $\rho$ is related to the sample size $n$; in our population setting with $n = \infty$, we thus take $\rho = \infty$. For the same reason, the resetting criterion (Line 9 in Algorithm E.2) reads $|S_2| < \infty$. In population setting, this means that the algorithm restarts only if $S_2 = \emptyset$.

Definition (E.5) contains two modifications w.r.t. the original (3)-(4). First, we consider only samples with bounded norm $\|x_i\|^2 \leq R$. Otherwise, with small probability, a sample $x_i$ may have large magnitude $\|x_i\|$ and consequently have large residual $r_{i,1}$, even if the estimate $\beta_1$ is close to the true $\beta_1^*$. Second, to homogenize the definitions of $S_1'$ and $S_2'$, we added the condition $w_{i,2} \leq w_{\text{th}}$ to the definition of $S_1'$ (E.5b), where $w_{i,2} = 1/(1 + \eta(x_i^\top \beta_2^{\text{phase-I}} - y_i)^2 / R)$ is calculated based on $\beta_2^{\text{phase-I}}$.

**Algorithm E.2:** `Mix-IRLS`: modified main phase for analysis purposes

---

**input** : samples $\{(x_i, y_i)\}_{i=1}^n$, parameters $\eta, w_{\text{th}}, R$

**output** : estimates $\beta_1^{(\text{phase-I})}, \beta_2^{(\text{phase-I})}$

**1** set $S_1 = [n]$

**2** initialize $\beta_1$ randomly

**3** compute $r_{i,1} = |x_i^\top \beta_1 - y_i|, \quad \forall i \in S_1$

**4** compute $w_{i,1} = (1 + \eta r_{i,1}^2/R)^{-1}, \quad \forall i \in S_1$

**5** compute $\beta_1 = (X_{S_1}^\top W_1 X_{S_1})^{-1} X_{S_1}^\top W_1 \, y_{S_1} \quad$ // $W_1 = \text{diag}(w_{1,1}, w_{2,1}, \ldots)$

**6** compute $r_{i,1} = |x_i^\top \beta_1 - y_i|, \quad \forall i \in S_1$

**7** compute $w_{i,1} = (1 + \eta r_{i,1}^2/R)^{-1}, \quad \forall i \in S_1$

**8** set $S_2 = S_2' = \{i \in S_1 : \|x_i\|^2 \leq R \text{ and } w_{i,1} \leq w_{\text{th}}\}$

**9** **if** $|S_2| < \infty$ **then**

**10** $\quad$ start `Mix-IRLS` over with $w_{\text{th}} \leftarrow w_{\text{th}} + 0.1$

**11** **end**

**12** compute $\beta_2^{(\text{phase-I})} = (X_{S_2'}^\top X_{S_2'})^{-1} X_{S_2'}^\top y_{S_2'}$

**13** compute $r_{i,2} = |x_i^\top \beta_2^{(\text{phase-I})} - y_i|, \quad \forall i \in S_1$

**14** compute $w_{i,2} = (1 + \eta r_{i,2}^2/R)^{-1}, \quad \forall i \in S_1$

**15** set $S_1' = \{i \in S_1 : \|x_i\|^2 \leq R \text{ and } w_{i,2} \leq w_{\text{th}}\}$

**16** compute $\beta_1^{(\text{phase-I})} = (X_{S_1'}^\top X_{S_1'})^{-1} X_{S_1'}^\top y_{S_1'}$

---

103  Theorem 6.1 is formulated in the large-$R$ regime. The following lemma is similar to Theorem 6.1,
104  but with the exact dependence on $R$. With this lemma in hand, Theorem 6.1 immediately follows. In
105  this section, we use the following notation for convenience:

$$s = \frac{\sigma_\epsilon}{\|\Delta\|} \quad \text{and} \quad \tilde{s} = \frac{s}{\sqrt{R}}. \tag{E.6}$$

106  **Lemma E.1.** *Let $\{(x_i, y_i)\}_{i=1}^\infty$, $K$, $p_1, p_2$, $\beta_1^*, \beta_2^*$, $\gamma$ and $q$ be defined as in Theorem 6.1. Let $(\beta_1, \beta_2)$*
107  *be an arbitrary initialization to `Mix-IRLS`, and denote $D = \|\beta_1 - \beta_1^*\|/\|\Delta\|$. Assume the parameters*
108  *of `Mix-IRLS` satisfy (12), $\rho = \infty$, and*

$$R > \max\left\{ \frac{1}{(q-\tilde{s})^2\|\Delta\|^2}, 5(3 \cdot \max\{1 + D, 3/2\} + s)^2\|\Delta\|^2\eta \right\}. \tag{E.7}$$

109  *Then the first phase of `Mix-IRLS` with at least one iteration ($T_1 \geq 1$) approximately recovers $\beta^*$,*

$$\max_{k=1,2} \|\beta_k - \beta_k^*\| < \frac{1}{\sqrt{R}} \frac{s}{s+\gamma}. \tag{E.8}$$

110  *Specifically, in the absence of noise ($s = 0$), `Mix-IRLS` perfectly recovers the two components,*
111  *$\beta_k = \beta_k^*$ for $k = 1, 2$.*

112  *Proof of Theorem 6.1.* The theorem follows by taking large enough $R$ in Lemma E.1. $\qquad\square$

### 113  E.1   Proof of Lemma E.1 and Proposition D.1

114  To prove Lemma E.1, we will use the following three auxiliary lemmas. Their proof appears in the
115  next subsections. In the following, unless otherwise stated, expectations are taken over all the random
116  variables (typically $x_i$, $\epsilon_i$ and $c_i^*$).

117  **Lemma E.2.** *Let $x \sim \mathcal{N}(0, I_d)$ and $\epsilon$ be independent random variables. Suppose $\epsilon$ has zero mean*
118  *with a symmetric distribution, $\mathbb{P}[\epsilon] = \mathbb{P}[-\epsilon]$. Let $u \in \mathbb{R}^d$ be a fixed unit vector, $\|u\| = 1$, and denote*
119  *$x_u = u^\top x$. Denote the events*

$$\mathcal{E} = \{|x_u| \leq t_1\}, \tag{E.9a}$$

$$S = \{\|x\|^2 \leq R \text{ and } (x_u - \epsilon)^2 \geq t_2\}, \tag{E.9b}$$

120 *for some fixed positive scalars $t_1, t_2, R$. Let $U = uu^\top$ and $x_\perp = x - x_u u$. Then*

$$\mathbb{E}\left[xx^\top \mid \mathcal{E}\right] = I_d - \left(1 - \mathbb{E}\left[x_u^2 \mid \mathcal{E}\right]\right)U, \tag{E.10}$$

$$\mathbb{E}\left[xx^\top \mid S\right]^{-1} = \frac{1}{\mathbb{E}[\|x_\perp\|^2 \mid S]}\left(I_d - U\right) + \frac{1}{\mathbb{E}[x_u^2 \mid S]}U. \tag{E.11}$$

**Lemma E.3.** *Assume the conditions of Lemma E.1. Denote $\tilde{\eta} = \eta\|\Delta\|^2/R$. Let $\beta_1^{(t)}$ be the $t$-th iterate of (2c) for the first component ($k = 1$), and denote $D_t = \|\beta_1^{(t)} - \beta_1^*\|/\|\Delta\|$. Then*

$$D_{t+1} \le \frac{5(q - \tilde{s})}{6}\left(1 + \tilde{\eta}(3(1 + D_t) + s)^2\right). \tag{E.12}$$

**Lemma E.4.** *Let $u, \Delta \in \mathbb{R}^d$, $\eta, R > 0$ and $q \in (0, 1/2)$ be fixed. Let $x \sim \mathcal{N}(0, I_d)$ and $\epsilon$ be independent random variables. Suppose $\epsilon$ is bounded, $|\epsilon| < s\|\Delta\|$ where $s < q\sqrt{R}$. Denote $w(x, \epsilon) = \left(1 + \eta\left(x^\top u + \epsilon\right)^2/R\right)^{-1}$. Further denote*

$$P = \mathbb{P}\left[w(x, \epsilon) < w_{th} \mid \|x\|^2 \le R\right] \tag{E.13}$$

*where $w_{th}$ satisfies (12). Then $P = 0$ if $\|u\|/\|\Delta\| \le q - \tilde{s}$ and $P > 0$ if $\|u\|/\|\Delta\| \ge 1 - q + \tilde{s}$.*

Let us briefly sketch the proof idea before we present it formally. Lemma E.2 is a technical result, used occasionally throughout the proof. Using Lemma E.3, we show that $\beta_1$ of Line 5 in Algorithm E.2 is a good approximation for the regression vector $\beta_1^*$. As a result, any sample with bounded norm that was generated from the first component has a small residual, and thus a large weight (Line 7 in Algorithm E.2). By removing all the samples with large and moderate weights (i.e., constructing the set $S_2$, Line 8), we are left with active samples from the second component only, as follows by Lemma E.4. Thus, $\beta_2^{\text{phase-I}}$ of Line 12 accurately estimates the regression vector $\beta_2^*$. Then, we similarly show that $\beta_1^{\text{phase-I}}$ of Line 16 accurately estimates the first vector $\beta_1^*$ as well. It is worth mentioning that due to the assumed imbalance, our method will indeed find $\beta_1^*$ as its first component and $\beta_2^*$ as its second.

*Proof of Lemma E.1.* As in Lemma E.3, let $\beta_1^{(t)}$ be the $t$-th iterate of (2c) for the first component ($k = 1$), and denote $D_t = \|\beta_1^{(t)} - \beta_1^*\|/\|\Delta\|$. In particular, $D_0 = D$. We shall prove by induction that for any $t \ge 1$, with $s$ and $\tilde{s}$ defined in (E.6),

$$D_t < q - \tilde{s} \le \gamma + s. \tag{E.14}$$

Combined with the assumption $q < 1/2$, the first inequality in (E.14) implies that $\beta_1^{(t)}$ is closer to $\beta_1^*$ than to $\beta_2^*$. As a consequence, after removal of samples with good to moderate fit, the remaining (poor fit) samples are all belong to the second component, as we prove below.

Let $t = 1$, and denote $\tilde{\eta} = \eta\|\Delta\|^2/R$. By Lemma E.3, after one iteration of the IRLS scheme ((2), or Lines 3 to 5 in Algorithm E.2), we have

$$D_1 \le \frac{5(q - \tilde{s})}{6}\left(1 + \tilde{\eta}(3(1 + D_0) + s)^2\right) < \frac{5(q - \tilde{s})}{6}\left(1 + \frac{1}{5}\right) = q - \tilde{s},$$

where in the second inequality we used $\tilde{\eta} \le (1/5)/(3(1 + D_0) + s)^2$, see (E.7). This proves (E.14) at $t = 1$. For the induction step, suppose (E.14) holds for some $t \ge 1$, namely $D_t < q - \tilde{s} < 1/2$. Recall that $q > \tilde{s}$. Invoking Lemma E.3 again yields

$$D_{t+1} \le \frac{5(q - \tilde{s})}{6}\left(1 + \tilde{\eta}(3(1 + D_t) + s)^2\right) < \frac{5(q - \tilde{s})}{6}\left(1 + \tilde{\eta}(9/2 + s)^2\right) \le \frac{5(q - \tilde{s})}{6}\left(1 + \frac{1}{5}\right)$$
$$= q - \tilde{s},$$

where in the last inequality we used $\tilde{\eta} \le (1/5)/(9/2 + s)^2$, see (E.7). This proves (E.14).

Applying (E.14) for $t = T_1 \ge 1$ yields

$$\frac{\|\beta_1^{(T_1)} - \beta_1^*\|}{\|\Delta\|} = D_{T_1} < q - \tilde{s}, \tag{E.15a}$$

$$\frac{\|\beta_1^{(T_1)} - \beta_2^*\|}{\|\Delta\|} \ge \frac{\|\beta_1^* - \beta_2^*\|}{\|\Delta\|} - \frac{\|\beta_1^{(T_1)} - \beta_1^*\|}{\|\Delta\|} = 1 - D_{T_1} > 1 - q + \tilde{s}. \tag{E.15b}$$

Hence, for any sample $x$ with $\|x\|^2 \le R$ and whose response belongs to the second component $(c^* = 2)$, the corresponding weight $w_1$ ((E.4), or Line 7 in Algorithm E.2) satisfies

$$\mathbb{P}\left[w_1 < w_{\text{th}} \mid c^* = 2 \text{ and } \|x\|^2 \le R\right] = \mathbb{P}\left[\frac{1}{1 + \eta\left(x^\top \beta_1^{(T_1)} - x^\top \beta_2^* - \epsilon\right)^2/R} < w_{\text{th}} \mid \|x\|^2 \le R\right] > 0,$$

as follows by combining (E.15b) and Lemma E.4 with $u = \beta_1^{(T_1)} - \beta_2^*$. In contrast, combining (E.15a) with the same lemma for $u = \beta_1^{(T_1)} - \beta_1^*$, gives that for any sample $x$ whose response belongs to the first component $(c^* = 1)$,

$$\mathbb{P}\left[w_1 < w_{\text{th}} \mid c^* = 1 \text{ and } \|x\|^2 \le R\right] = \mathbb{P}\left[\frac{1}{1 + \eta\left(x^\top \beta_1^{(T_1)} - x^\top \beta_1^* - \epsilon\right)^2/R} < w_{\text{th}} \mid \|x\|^2 \le R\right] = 0.$$

Hence, by (E.5a), all the (infinite number of) samples in $S_2$ belong to the second component $c^* = 2$. In other words, the choice of the threshold $w_{\text{th}}$ allows to detect a subset of samples $(x, y)$ that all belong to the second component. Note that the resetting criterion $|S_2| < \infty$ (Line 9 in Algorithm E.2) does not hold, as $S_2$ is an infinite set.

Since all the samples in $S_2 = S_2'$ belong to the second component, their responses follow the relation $y = x^\top \beta_2^* + \epsilon$. The final estimate of the first phase for the second component ((5), or Line 12 in Algorithm E.2) is thus

$$\beta_2^{\text{phase-I}} = \mathbb{E}\left[xx^\top \mid S_2\right]^{-1} \mathbb{E}\left[xy \mid S_2\right] = \mathbb{E}\left[xx^\top \mid S_2\right]^{-1} \mathbb{E}\left[x(x^\top \beta_2^* + \epsilon) \mid S_2\right]$$
$$= \beta_2^* + \mathbb{E}\left[xx^\top \mid S_2\right]^{-1} \mathbb{E}\left[\epsilon \cdot x \mid S_2\right],$$

as follows by the weak law of large numbers. Rearranging and taking the norm of both sides gives

$$\left\|\beta_2^{\text{phase-I}} - \beta_2^*\right\| = \left\|\mathbb{E}\left[xx^\top \mid S_2\right]^{-1} \mathbb{E}\left[\epsilon \cdot x \mid S_2\right]\right\|. \tag{E.16}$$

To upper bound the RHS of (E.16), we shall analyze each of the two terms $\mathbb{E}\left[xx^\top \mid S_2\right]^{-1}$ and $\mathbb{E}\left[\epsilon \cdot x \mid S_2\right]$. Let $u = \beta_1^{(T_1)} - \beta_1^*$, and decompose $x = x_u \tilde{u} + x_\perp$ where $u \perp x_\perp$ and $\tilde{u} = u/\|u\|$. Invoking Lemma E.2 implies that the first term satisfies

$$\mathbb{E}\left[xx^\top \mid S_2\right]^{-1} = \frac{1}{\mathbb{E}\left[\|x_\perp\|^2 \mid S_2\right]}\left(I_d - \tilde{u}\tilde{u}^\top\right) + \frac{1}{\mathbb{E}\left[x_u^2 \mid S_2\right]}\tilde{u}\tilde{u}^\top.$$

To analyze the second term on the RHS of (E.16), recall the definition of $S_2$ in (E.5a). The weight condition of $S_2$, $w_2 \le w_{\text{th}}$, involves only the $x_u$ part of $x$. Together with the isotropic distribution assumption on $x$ (9), it follows that $x_\perp$ is isotropically distributed even when conditioned on $S_2$. Hence, the second term on the RHS of (E.16) satisfies

$$\mathbb{E}\left[\epsilon \cdot x \mid S_2\right] = \mathbb{E}\left[\epsilon \cdot x_u \mid S_2\right]\tilde{u}.$$

Inserting these two equalities into (E.16) yields

$$\left\|\beta_2^{\text{phase-I}} - \beta_2^*\right\| = \frac{|\mathbb{E}\left[\epsilon \cdot x_u \mid S_2\right]|}{\mathbb{E}\left[x_u^2 \mid S_2\right]} \le s\|\Delta\| \cdot \frac{\mathbb{E}\left[|x_u| \mid S_2\right]}{\mathbb{E}\left[x_u^2 \mid S_2\right]} \le \frac{s\|\Delta\|}{\sqrt{\mathbb{E}\left[x_u^2 \mid S_2\right]}}, \tag{E.17}$$

where the first inequality follows by the bounded noise assumption (10), and the second by Jensen's inequality $\mathbb{E}[|x_u|]^2 \le \mathbb{E}[x_u^2]$. We shall now lower bound $\mathbb{E}\left[x_u^2 \mid S_2\right]$. For any pair $(x, y) \in S_2$, the weight satisfies

$$\frac{1}{1 + \eta\left(x_u - \epsilon\right)^2/R} = w_1 \le w_{\text{th}} < \frac{1}{1 + \eta q^2\|\Delta\|^2},$$

where the second inequality follows by (12). Rearranging and taking the square root gives that

$$|x_u - \epsilon| > q\|\Delta\|\sqrt{R}.$$

By the triangle inequality,

$$|x_u| > q\|\Delta\|\sqrt{R} - |\epsilon| \overset{(a)}{\geq} (q\sqrt{R} - s)\|\Delta\| \overset{(b)}{=} (\gamma + s)\|\Delta\|\sqrt{R},$$

where (a) follows by combining (10) and (E.6), and (b) by the definition of $q$ (11). In particular, $\mathbb{E}\left[x_u^2 \mid S_2\right] > (\gamma + s)^2 \|\Delta\|^2 R$. Plugging this into (E.17) yields

$$\left\|\beta_2^{\text{phase-I}} - \beta_2^*\right\| < \frac{1}{\sqrt{R}}\frac{s}{\gamma + s}. \tag{E.18}$$

This completes the analysis of the second round of the IRLS scheme, and proves (E.8) at $k = 2$.

Finally, we need to prove (E.8) at $k = 1$, by deriving a similar bound for the estimate $\beta_1^{\text{phase-I}}$ of the first component. Dividing (E.18) by $\|\Delta\|$ gives

$$\frac{\left\|\beta_2^{\text{phase-I}} - \beta_2^*\right\|}{\|\Delta\|} < \frac{1}{\|\Delta\|\sqrt{R}}\frac{s}{\gamma + s} \leq \frac{1}{\|\Delta\|\sqrt{R}} < q - \tilde{s},$$

where the last inequality follows by (E.7). This bound is identical to (E.15a), but now for the accuracy of the second component rather than the first one. Since the subset $S_1'$, which is calculated using $\beta_2^{\text{phase-I}}$, is defined similarly to $S_2'$ (see (E.5), or Lines 8 and 15 in Algorithm E.2), the rest of the argument follows the lines of the second component analysis described above, and we omit its details.

$\square$

*Proof of Proposition D.1.* For any $i \in S_2$,

$$w_{i,2} = \frac{1}{1 + \eta(x_i^\top(\beta_2^{\text{phase-I}} - \beta_2^*) + \epsilon_i)^2/R} \geq \frac{1}{1 + \eta\left(\|x_i\|/\sqrt{R} + s\|\Delta\|\right)^2/R}$$

$$\geq \frac{1}{1 + \eta\left(1 + s\|\Delta\|\right)^2/R},$$

where the first inequality follows by (E.18) and (10), and the second by the condition $\|x_i\|^2 \leq R$ in the definition of $S_2$. For large enough $R$, we get

$$w_{i,2} \geq \frac{1}{1 + \eta q^2\|\Delta\|^2} > w_{\text{th}},$$

where the second inequality follows by (12). As a result, $S_3 = \emptyset$ according to the definition of $S_3$ in (3).

The second part of the proposition follows since the estimators are identical in the known and unknown $K$ settings. $\square$

## E.2 Proof of Lemma E.2

*Proof of Lemma E.2.* Given that $x = x_u u + x_\perp$ and $U = uu^\top$,

$$xx^\top = x_u^2 U + x_u u x_\perp^\top + x_u x_\perp u^\top + x_\perp x_\perp^\top.$$

First, let us show that the two middle terms, $x_u u x_\perp^\top$ and $x_u x_\perp u^\top$, vanish in expectation conditional on either $\mathcal{E}$ or $S$. The case of $\mathcal{E}$ is simpler: as $x_\perp$ is independent of $x_u$ and thus also independent of $\mathcal{E}$,

$$\mathbb{E}\left[x_u x_\perp u^\top \mid \mathcal{E}\right] = \mathbb{E}[x_u \mid \mathcal{E}] \cdot \mathbb{E}[x_\perp]u^\top = 0.$$

Similarly, $\mathbb{E}\left[x_u u x_\perp^\top \mid \mathcal{E}\right] = 0$ as well.

Next, let us analyze $\mathbb{E}\left[x_u x_\perp u^\top \mid S\right]$. Unconditioned on $S$, $x_u$ is normally distributed around zero, and in particular symmetric. In addition, unconditioned on $S$, $\epsilon$ is symmetric and independent of $x_u$.

201 Hence, $\mathbb{P}\left[(x_u, \epsilon)\right] = \mathbb{P}[x_u] \cdot \mathbb{P}[\epsilon] = \mathbb{P}[-x_u] \cdot \mathbb{P}[-\epsilon] = \mathbb{P}[(-x_u, -\epsilon)]$. Together with Bayes' theorem

202 and the fact that $\mathbb{P}\left[S \mid (x_u, \epsilon)\right] = \mathbb{P}[S \mid (-x_u, -\epsilon)]$, we conclude

$$\mathbb{P}[(x_u, \epsilon) \mid S] = \frac{\mathbb{P}[S \mid (x_u, \epsilon)] \cdot \mathbb{P}[(x_u, \epsilon)]}{\mathbb{P}[S]} = \frac{\mathbb{P}[S \mid (-x_u, -\epsilon)] \cdot \mathbb{P}[(-x_u, -\epsilon)]}{\mathbb{P}[S]}$$
$$= \mathbb{P}[(-x_u, -\epsilon) \mid S].$$

203 As a result, the marginal distribution of $x_u$ conditional on $S$ is also symmetric,

$$\mathbb{P}[x_u \mid S] = \mathbb{E}_\epsilon[\mathbb{P}[(x_u, \epsilon) \mid S]] = \mathbb{E}_\epsilon[\mathbb{P}[(-x_u, -\epsilon) \mid S]] = \mathbb{E}_\epsilon[\mathbb{P}[(-x_u, \epsilon) \mid S]]$$
$$= \mathbb{P}[-x_u \mid S].$$

204 Further, $x_u$ and $x_\perp$ are independent when unconditioned on $S$, and the coupling between $x_u$ and

205 $x_\perp$ under the event $S$ is only by the inequality $\|x_\perp\|^2 \le R - x_u^2$, namely it depends only on the

206 magnitudes $|x_u|$ and $\|x_\perp\|$. Hence,

$$\mathbb{P}[(x_u, x_\perp) \mid S] = \frac{\mathbb{P}[S \mid (x_u, x_\perp)] \cdot \mathbb{P}[(x_u, x_\perp)]}{\mathbb{P}[S]} = \frac{\mathbb{P}[S \mid (x_u, x_\perp)] \cdot \mathbb{P}[x_u] \cdot \mathbb{P}[x_\perp]]}{\mathbb{P}[S]}$$
$$= \frac{\mathbb{P}[S \mid (-x_u, x_\perp)] \cdot \mathbb{P}[-x_u] \cdot \mathbb{P}[x_\perp]}{\mathbb{P}[S]} = \frac{\mathbb{P}[S \mid (-x_u, x_\perp)] \cdot \mathbb{P}[(-x_u, x_\perp)]}{\mathbb{P}[S]}$$
$$= \mathbb{P}[(-x_u, x_\perp) \mid S].$$

207 This implies that $\mathbb{E}[x_u x_\perp \mid S] = 0$, so that $\mathbb{E}[x_u x_\perp u^\top \mid S] = 0$. Similarly, $\mathbb{E}[x_u u x_\perp^\top \mid S] = 0$.

208 Next, we analyze the last term $x_\perp x_\perp^\top$. Again, the case of $\mathcal{E}$ is simple: since $x_\perp$ is independent of $\mathcal{E}$,

209 $\mathbb{E}\left[x_\perp x_\perp^\top \mid \mathcal{E}\right] = \mathbb{E}\left[x_\perp x_\perp^\top\right] = I_d - U$. This proves (E.10).

210 Finally, to prove (E.11), we analyze the case of $S$. Let $e_i$ be the $i$-th standard basis vector. W.l.o.g.,

211 assume $u = e_d = (0, \ldots, 0, 1)^\top$. Decompose $x_\perp = \sum_{i=1}^{d-1} a_i e_i$. Recall that conditional

212 on the event $S$, $x_\perp$ still has a spherically symmetric distribution. In particular, for any value of

213 $\|x_\perp\| = t$, the vector $x_\perp$ is uniformly distributed on the sphere of radius $t$. Hence, for any $i \ne j$,

214 $\mathbb{E}[a_i a_j \mid S] = \mathbb{E}_t \left[\mathbb{E}[a_i a_j \mid S, \|x_\perp\| = t]\right]$ vanishes, which implies

$$\mathbb{E}[x_\perp x_\perp^\top \mid S] = \mathbb{E}\left[\sum_{i,j=1}^{d-1} a_i a_j e_i e_j^\top\right] = \sum_{i=1}^{d-1} \mathbb{E}[a_i^2] e_i e_i^\top$$
$$= \mathbb{E}[a_1^2 \mid S] \sum_{i=1}^{d-1} e_i e_i^\top = \mathbb{E}[a_1^2 \mid S](I_d - U).$$

215 Let $\alpha_u = \mathbb{E}[x_u^2 \mid S]$ and $\alpha_\perp = \mathbb{E}[a_1^2 \mid S]$. Then

$$\mathbb{E}[xx^\top \mid S] = \alpha_u U + \alpha_\perp (I_d - U) = \alpha_\perp \left(I_d - \frac{\alpha_\perp - \alpha_u}{\alpha_\perp} U\right).$$

216 Its inverse is

$$\mathbb{E}[xx^\top \mid S]^{-1} = \frac{1}{\alpha_\perp}\left(I_d + \frac{\alpha_\perp - \alpha_u}{\alpha_u} U\right) = \frac{1}{\alpha_\perp}(I_d - U) + \frac{1}{\alpha_u} U.$$

217 $\qquad\qquad\qquad\qquad\qquad\qquad\qquad\qquad\qquad\qquad\qquad\qquad\qquad\qquad\qquad\qquad\qquad\qquad\qquad\quad$ $\square$

## E.3  Proof of Lemma E.3

219 To prove Lemma E.3, we state and prove the following auxiliary result. In this subsection, we use

220 the notation $A \succeq B$ to indicate that a pair of matrices $A, B$ satisfies the semidefinite positive cone

221 inequality, namely $A - B$ is positive semidefinite.

222 **Lemma E.5.** *Let $u \in \mathbb{R}^d$ and $x \sim \mathcal{N}(0, I_d)$. Let $z \in \mathbb{R}$ be a bounded random variable, $|z| \le z_{max}$,*

223 *independent of $x$. Denote $w(x, z) = \left(1 + \left(x^\top u + z\right)^2\right)^{-1}$. Then*

$$\|\mathbb{E}\left[w(x, z) \cdot zx\right]\| \le \sqrt{\frac{2}{\pi}} z_{max}, \tag{E.19}$$

224 *and*

$$\frac{24/25}{1 + (3\|u\| + z_{max})^2} \leq \sigma_{min}\left(\mathbb{E}\left[w(x,z) \cdot xx^\top\right]\right) \leq \left\|\mathbb{E}\left[w(x,z) \cdot xx^\top\right]\right\| \leq 1. \qquad (E.20)$$

225 *Proof.* If $u = 0$ then the lemma holds, as in this case $w = 1/(1 + z^2) \in (0,1]$, it is independent of $x$,
226 and $\mathbb{E}[x] = 0$. Hence, we assume $u \neq 0$. Decompose $x = x_u \tilde{u} + x_\perp$ where $u \perp x_\perp$ and $\tilde{u} = u/\|u\|$.
227 Since $x \sim \mathcal{N}(0, I_d)$, we have $x_u \sim \mathcal{N}(0,1)$ and $x_\perp \sim \mathcal{N}(0, I_d - \tilde{u}\tilde{u}^\top)$. Furthermore, $x_u$ and $x_\perp$
228 are independent, and $\mathbb{E}[x_\perp] = 0$. Hence,

$$\|\mathbb{E}\left[w(x,z) \cdot zx\right]\| = \|\mathbb{E}\left[w(x_u\tilde{u}, z) \cdot z(x_u\tilde{u} + x_\perp)\right]\| = |\mathbb{E}\left[w(x_u\tilde{u}, z) \cdot zx_u\right]| + \|\mathbb{E}\left[w(x_u\tilde{u}, z) \cdot z\right] \cdot \mathbb{E}[x_\perp]\|$$

$$\leq z_{\max} \cdot \mathbb{E}[|x_u|] = \sqrt{\frac{2}{\pi}} z_{\max},$$

229 where the inequality follows by $0 \leq w(x,z) \leq 1$ and $\mathbb{E}[x_\perp] = 0$. This proves (E.19).

230 Next, we prove the lower bound on $\sigma_{min}\left(\mathbb{E}\left[w(x,z) \cdot xx^\top\right]\right)$. Let $t > 0$, and consider the event
231 $\mathcal{E}_t = \{|x_u| \leq t\}$. Conditional on this event, $w(x,z) \geq 1/(1 + (t\|u\| + z_{\max})^2)$. Hence,

$$\mathbb{E}\left[w(x,z) \cdot xx^\top\right] \succeq \mathbb{P}[\mathcal{E}] \cdot \mathbb{E}\left[w(x,z) \cdot xx^\top \mid \mathcal{E}\right] \succeq \frac{\mathbb{P}[\mathcal{E}]}{1 + (t\|u\| + z_{\max})^2}\mathbb{E}\left[xx^\top \mid \mathcal{E}\right]. \qquad (E.21)$$

232 It is left to lower bound $\mathbb{E}\left[xx^\top \mid \mathcal{E}\right]$. Let $U = \tilde{u}\tilde{u}^\top$. Since $x_\perp$ is independent of $x_u$, it is also
233 independent of $\mathcal{E}$. Lemma E.2 thus implies

$$\mathbb{E}\left[xx^\top \mid \mathcal{E}\right] = I_d - \left(1 - \mathbb{E}\left[x_u^2 \mid \mathcal{E}\right]\right)U.$$

234 Since $x_u \sim \mathcal{N}(0,1)$, we have

$$\mathbb{E}\left[x_u^2 \mid \mathcal{E}\right] = \frac{\frac{1}{\sqrt{2\pi}}\int_{-t}^{t} x_u^2 e^{-x_u^2/2}dx_u}{\mathbb{P}[\mathcal{E}]} = 1 - \sqrt{\frac{2}{\pi}}\frac{t \cdot e^{-\frac{t^2}{2}}}{\mathbb{P}[\mathcal{E}]}.$$

235 Hence,

$$\mathbb{E}\left[xx^\top \mid \mathcal{E}\right] = I_d - \sqrt{\frac{2}{\pi}}\frac{t \cdot e^{-\frac{t^2}{2}}}{\mathbb{P}[\mathcal{E}]}U.$$

236 Plugging this equality into (E.21) and using $\mathbb{P}[\mathcal{E}] = \mathbb{P}[|x_u| \leq t] = 2\Phi(t) - 1$ gives

$$\mathbb{E}\left[w(x,z) \cdot xx^\top \mid \mathcal{E}\right] \succeq \frac{1}{1 + (t\|u\| + z_{\max})^2}\left((2\Phi(t) - 1) \cdot I_d - \sqrt{\frac{2}{\pi}}t \cdot e^{-\frac{t^2}{2}} \cdot U\right).$$

237 The RHS is, up to scaling, a rank one perturbation of the identity matrix. Hence, its smallest singular
238 value is

$$\sigma_{min}\left((2\Phi(t) - 1)I_d - t\sqrt{\frac{2}{\pi}}e^{-t^2/2}U\right) = 2\Phi(t) - 1 - t\sqrt{\frac{2}{\pi}}e^{-t^2/2}.$$

239 The lower bound in (E.20) follows by picking $t = 3$, as $2\Phi(3) - 1 - 3\sqrt{\frac{2}{\pi}}e^{-9/2} > 24/25$.

240 Finally, the upper bound in (E.20) follows trivially by $0 \leq w(x,z) \leq 1$ and $\mathbb{E}\left[xx^\top\right] = I_d$.

241 $\qquad \qquad \qquad \qquad \qquad \qquad \qquad \qquad \qquad \qquad \qquad \qquad \qquad \qquad \qquad \qquad \qquad \qquad \qquad \square$

242 *Proof of Lemma E.3.* Fix some $t$, and denote the iterates at time steps $t$ and $(t+1)$ by $\beta_1 = \beta_1^{(t)}$ and
243 $\beta_1^+ = \beta_1^{(t+1)}$, respectively. Further denote $D = D_t$. According to (2c),

$$\beta_1^+ = \left(\sum_{i=1}^{n} w_{i,1}x_i x_i^\top\right)^{-1}\left(\sum_{i=1}^{n} w_{i,1}x_i y_i\right).$$

244 Recall that the cluster assignment of a sample, $c_i^* \in \{1,2\}$, is distributed as Bernoulli with
245 probabilities $p_1, p_2$, independently of the sample $x_i$. Let $r(x, c^*, \epsilon; \beta_1) = |x^\top\beta_1 - y(c^*, \epsilon)|$ and

$246$  $w(x, c^*, \epsilon; \beta_1) = 1/\left(1 + \eta r(x, c^*, \epsilon; \beta_1)^2/R\right)$ where $y(c^*, \epsilon) = x^\top \beta_{c^*}^* + \epsilon$. For simplicity of
$247$  notation, from now on we suppress the dependencies of $r, w$ and $y$ on $x, c^*, \epsilon$ and $\beta_1$.

$248$  By the weak law of large numbers, as $n \to \infty$, the terms $\left(\sum_{i=1}^n w_i x_i x_i^\top\right)^{-1}$ and $\sum_{i=1}^n w_i x_i y_i$
$249$  converge to $\mathbb{E}\left[w \cdot xx^\top\right]^{-1}$ and $\mathbb{E}[w \cdot xy]$, respectively, with the expectation taken over $x, c^*$ and $\epsilon$.
$250$  Hence, in the population setting, the update of $\beta_1$ takes the form

$$\beta_1^+ \overset{n \to \infty}{\to} \mathbb{E}\left[w \cdot xx^\top\right]^{-1} \mathbb{E}[w \cdot xy].$$

$251$  Observe that the true regression vector $\beta_1^*$ can be written as

$$\beta_1^* = \left(\mathbb{E}\left[w \cdot xx^\top\right]\right)^{-1} \mathbb{E}\left[w \cdot xx^\top\right] \beta_1^* = \left(\mathbb{E}\left[w \cdot xx^\top\right]\right)^{-1} \mathbb{E}\left[w \cdot xx^\top \beta_1^*\right].$$

$252$  Hence, the distance of the next iterate $\beta_1^+$ from $\beta_1^*$ satisfies

$$\|\beta_1^+ - \beta_1^*\| = \left\|\left(\mathbb{E}\left[w \cdot xx^\top\right]\right)^{-1} \mathbb{E}\left[w \cdot x(y - x^\top \beta_1^*)\right]\right\| \leq \frac{\left\|\mathbb{E}\left[w \cdot x(y - x^\top \beta_1^*)\right]\right\|}{\sigma_{\min}\left(\mathbb{E}\left[w \cdot xx^\top\right]\right)}. \quad \text{(E.22)}$$

$253$  We first analyze the denominator of the RHS in (E.22). Denote $\delta = \beta_1 - \beta_1^*$, and for $k = 1, 2$ let
$254$  $w_{c^*=k}$ be the weight conditional on the response $y$ having been generated from the $k$-th component.
$255$  By the independence of $c^*$ from $x$ and $\epsilon$,

$$\mathbb{E}\left[w \cdot xx^\top\right] = p_1 \mathbb{E}\left[w_{c^*=1} \cdot xx^\top\right] + p_2 \mathbb{E}\left[w_{c^*=2} \cdot xx^\top\right]. \quad \text{(E.23)}$$

$256$  Conditional on the response $y$ having been generated from the first component ($c^* = 1$), the weight
$257$  satisfies

$$w_{c^*=1} = \frac{1}{1 + \eta(x^\top \beta_1 - x^\top \beta_1^* - \epsilon)^2/R} = \frac{1}{1 + \eta(x^\top \delta - \epsilon)^2/R}.$$

$258$  Invoking Eq. (E.20) of Lemma E.5 with $u = \sqrt{\eta/R}\delta$, $z = \sqrt{\eta/R}\epsilon$ and $z_{\max} = \sqrt{\eta/R}s\|\Delta\|$ gives

$$\sigma_{\min}\left(\mathbb{E}\left[w_{c^*=1} \cdot xx^\top\right]\right) \geq \frac{24}{25} \frac{1}{1 + \eta(3\|\delta\| + s\|\Delta\|)^2/R} \geq \frac{24}{25} \frac{1}{1 + \tilde{\eta}(3D + s)^2}. \quad \text{(E.24)}$$

$259$  Similarly, conditional on the second component ($c^* = 2$), the weight satisfies

$$w_{c^*=2} = \frac{1}{1 + \eta(x^\top \beta_1 - x^\top \beta_2^* - \epsilon)^2/R} = \frac{1}{1 + \eta(x^\top(\delta + \Delta) - \epsilon)^2/R}.$$

$260$  Lemma E.5 thus implies

$$\sigma_{\min}\left(\mathbb{E}\left[w_{c^*=2} \cdot xx^\top\right]\right) \geq \frac{24}{25} \frac{1}{1 + \tilde{\eta}(3(1 + D) + s)^2}. \quad \text{(E.25)}$$

$261$  Plugging these inequalities into (E.23) yields

$$\sigma_{\min}\left(\mathbb{E}\left[w \cdot xx^\top\right]\right) \geq \frac{24p_1}{25} \frac{1}{1 + \tilde{\eta}(3D + s)^2} + \frac{24p_2}{25} \frac{1}{1 + \tilde{\eta}(3(1 + D) + s)^2}$$
$$\geq (p_1 + p_2) \frac{24}{25} \frac{1}{1 + \tilde{\eta}(3(1 + D) + s)^2} = \frac{24}{25} \frac{1}{1 + \tilde{\eta}(3(1 + D) + s)^2}. \quad \text{(E.26)}$$

$262$  Next, we upper bound the numerator of the RHS in (E.22). By the triangle inequality,

$$\left\|\mathbb{E}\left[w \cdot x(y - x^\top \beta_1^*)\right]\right\| = \left\|p_1 \mathbb{E}\left[w \cdot x(y - x^\top \beta_1^*) \mid c^* = 1\right] + p_2 \mathbb{E}[w \cdot x(y - x^\top \beta_1^*) \mid c^* = 2]\right\|$$
$$\leq p_1 \|\mathbb{E}\left[w_{c^*=1} \cdot \epsilon x\right]\| + p_2 \|\mathbb{E}\left[w_{c^*=2} \cdot x(\epsilon - x^\top \Delta)\right]\|$$
$$\leq p_1 \|\mathbb{E}\left[w_{c^*=1} \cdot \epsilon x\right]\| + p_2 \|\mathbb{E}\left[w_{c^*=2} \cdot \epsilon x\right]\| + p_2 \|\mathbb{E}\left[w_{c^*=2} \cdot xx^\top\right]\| \cdot \|\Delta\|.$$

$263$  We now employ Lemma E.5 to bound each of the three terms on the RHS. The first term is bounded
$264$  using (E.19) with $u = \sqrt{\eta/R}\delta$, $z = \sqrt{\eta/R}\epsilon$ and $z_{\max} = \sqrt{\eta/R}s\|\Delta\|$. The second term is bounded
$265$  using (E.19) with $u = \sqrt{\eta/R}(\delta + \Delta)$ and the same $z, z_{\max}$. The third term is bounded using (E.20)
$266$  with the same $u, z, z_{\max}$. Putting everything together, we obtain

$$\left\|\mathbb{E}\left[w \cdot x(y - x^\top \beta_1^*)\right]\right\| \leq \sqrt{\frac{2}{\pi}}(p_1 + p_2)s\|\Delta\| + p_2\|\Delta\| = \left(\sqrt{\frac{2}{\pi}}s + p_2\right)\|\Delta\|.$$

Plugging this, together with (E.26), into (E.22), gives

$$\|\beta_1^+ - \beta_1^*\| \leq \frac{25}{24}\left(\sqrt{\frac{2}{\pi}}s + p_2\right)\left(1 + \tilde{\eta}(3(1+D)+s)^2\right)\|\Delta\| \leq \frac{25}{24}\left(\frac{4s}{5} + p_2\right)\left(1 + \tilde{\eta}(3(1+D)+s)^2\right)\|\Delta\|$$

$$= \frac{5}{6}(q-\tilde{s})\left(1 + \tilde{\eta}(3(1+D)+s)^2\right)\|\Delta\|,$$

where in the last equality we used the definition $q = (5p_2 + 4s)/4 + \tilde{s}$ (11).

$\square$

## E.4 Proof of Lemma E.4

*Proof of Lemma E.4.* Suppose $\|u\|/\|\Delta\| \leq q - \tilde{s}$ and $\|x\|^2 \leq R$. Let $x_u = x^\top u/\|u\|$, so that $x^\top u = x_u\|u\|$. Recall that $\tilde{s} = s/\sqrt{R}$ (E.6). Since $q > \tilde{s}$,

$$\frac{|x^\top u + \epsilon|}{\|\Delta\|} = \frac{|x_u\|u\| + \epsilon|}{\|\Delta\|} \leq |x_u|(q-\tilde{s}) + s \leq \sqrt{R}(q-\tilde{s}) + s = \sqrt{R}q.$$

Hence,

$$w(x,\epsilon) \geq \frac{1}{1 + \eta q^2\|\Delta\|^2} > w_{\text{th}},$$

where the second inequality follows by (12). This proves the first part of the lemma.

Next, suppose $\|u\|/\|\Delta\| \geq 1 - q + \tilde{s}$ and $\|x\|^2 \leq R$. Let $\delta = 1 - |x_u|/\sqrt{R} \geq 0$. Then

$$\frac{|x^\top u + \epsilon|}{\|\Delta\|} = \frac{|x_u\|u\| + \epsilon|}{\|\Delta\|} \geq |x_u|(1 - q + \tilde{s}) - s$$

$$= \sqrt{R}(1-\delta)(1-q) - \delta s$$

$$= \sqrt{R}\left((1-\delta)(1-q) - \delta s/\sqrt{R}\right),$$

so that

$$w(x,\epsilon) \leq \frac{1}{1 + \eta\left((1-\delta)(1-q) - \delta s/\sqrt{R}\right)^2\|\Delta\|^2}.$$

The RHS is monotonically increasing in $\delta$. For $\delta = 0$, we get $w(x) \leq 1/\left(1 + \eta(1-q)^2\|\Delta\|^2\right)$. Hence, for any $\zeta > 0$, there exists a sufficiently small $\delta > 0$ such that

$$w(x,\epsilon) - \frac{1}{1 + \eta(1-q)^2\|\Delta\|^2} < \zeta.$$

Let $\zeta = w_{\text{th}} - 1/\left(1 + \eta(1-q)^2\|\Delta\|^2\right)$. By (12), $\zeta > 0$. Hence, there exists $\delta > 0$ such that

$$w(x,\epsilon) - \frac{1}{1 + \eta(1-q)^2\|\Delta\|^2} < w_{\text{th}} - \frac{1}{1 + \eta(1-q)^2\|\Delta\|^2},$$

or, equivalently, $w(x,\epsilon) < w_{\text{th}}$. The probability $P$ for this event satisfies

$$P = \mathbb{P}[1 - |x_u|/\sqrt{R} \leq \delta \mid \|x\|^2 \leq R] = \mathbb{P}[x_u^2 \geq (1-\delta)^2 R \mid \|x\|^2 \leq R] > 0.$$

$\square$

# F  Further discussion of `Mix-IRLS`'s mechanism

In Figure 1 of the main text, we pictorially illustrated the mechanism of `Mix-IRLS` given an imbalanced mixture. To complement this illustration, we ran a simulation and traced the evolution of `Mix-IRLS`'s first component estimate at various inner iterations (see Eq. (2) of the main text). The result is shown in Figure F.1. Interestingly, at the first iterations, the estimate of `Mix-IRLS` is

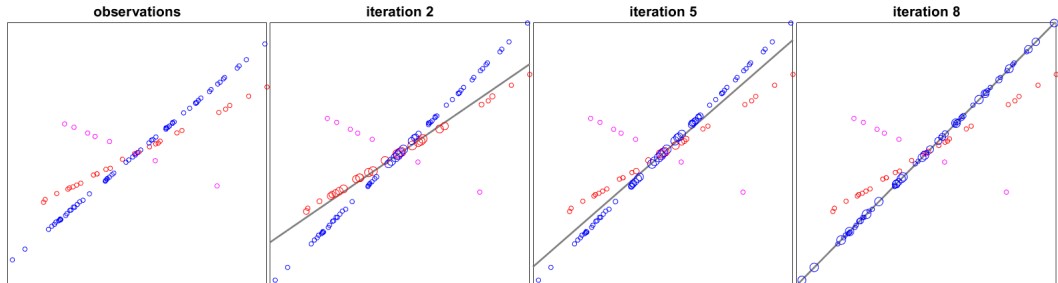

Figure F.1: Illustration of the evolution of `Mix-IRLS` at various iterations till finding the first mixture component, at the same setting as in Figure 2. Different colors correspond to different components, and marker size corresponds to the associated weight $w_{i,k=1}$; see (2b).

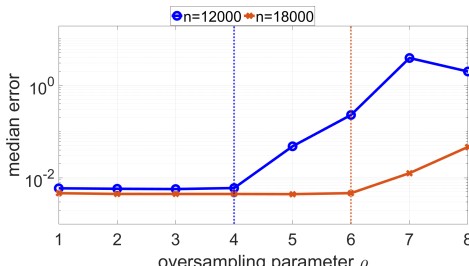

Figure F.2: the effect of the oversampling parameter $\rho$ on the performance of `Mix-IRLS`. Vertical lines correspond to the data oversampling ratio $n = n_{\text{info}}$. The setting in both panels are the same as in Figure 2, and the median is taken over 50 repetitions.

closer to the second component, whose proportion is only $p_2 = 0.2$. This must be a consequence of the random initialization, which happened to be closer to the second component. However, as the iterations proceed, `Mix-IRLS` is shown to gradually set its focus to the dominant component ($p_1 = 0.7$), as expected.

Next, we elaborate on the tuning-free nature of `Mix-IRLS` briefly discussed in Remark 3.2. Let us first focus on the oversampling parameter $\rho$. To estimate a component, we must use at least $d$ samples, otherwise the system is underdetermined (see Section 2). In our algorithm, we use (the largest-weighted) $\rho \cdot d$ samples; see Eq. (4). If the sample size is large enough, then $\rho$ has little effect: we can use many samples to estimate a component ($\rho \gg 1$), and still have enough samples for the next components. However, in extreme cases where the sample size is close to the information limit, $\rho$ should be close to 1, so that we use the minimal number of samples for each component. This idea is demonstrated in Figure F.2. It is shown that `Mix-IRLS` starts to perform worse when $\rho > n/n_{\text{info}}$, as expected. Empirically, also the other parameters of `Mix-IRLS` need not be carefully tuned: $T_1$ is simply the number of iteration, and only need to be large enough; $w_{\text{th}}$ is dynamically adapted throughout the algorithm as explained in Remark 3.1; and since the residual $r$ in (2b) is normalized by $\bar{r}$, the coefficient parameter $\eta$ is $\mathcal{O}(1)$, and in practice a fixed value for $\eta$ works well in a wide range of settings.

The first phase of `Mix-IRLS` involves a refinement step; see (5). In practice, the effect of this step is negligible. However, conceptually, it makes sense to use only the largest-weighted samples - namely, the samples that belong to the estimated component with high confidence - to re-estimate the component.

Our theoretical guarantee (Theorem 6.1) does not hold in the perfectly balanced case ($p_1 = p_2 = 1/2$). In practice, however, `Mix-IRLS` does succeed also in this case. The reason is that even if the mixture is perfectly balanced, the different sample magnitudes $\|x_i\|$ and noise terms $|\epsilon_i|$ make the problem asymmetric. As such, `Mix-IRLS` will gradually tend towards one of the components, and eventually recover it. Only in the population setting perfect balance is a symmetric pathology in which `Mix-IRLS` enjoys no guarantee.

## G  Additional Simulation and Experimental Details

All algorithms get as input a maximal number of iterations, and `Mix-IRLS:tuned` and GD have additional parameters. The maximal number of iterations in `Mix-IRLS`, `Mix-IRLS:tuned`, `AltMin` and `EM` was set to $10^3$, and to $10^5$ in GD. The parameter $\rho$ of `Mix-IRLS` and `Mix-IRLS:tuned` was fixed at the value of 1 in synthetic simulations and 2 in real-world experiments. To tune $\eta$ and $w_{\text{th}}$ of `Mix-IRLS:tuned` and the step size $\eta_{\text{GD}}$ of GD, we run each simulation and experimental setting with a different set of values, and choose the best values out of 10 repetitions. The allowed values were: $\eta = \sqrt{\Phi^{-1}(0.75)/\nu} = \sqrt{0.6745/\nu}$ where $\nu \in \{0.1, 0.5, 1, 2\}$, $w_{\text{th}} \in \{0.01, 0.1, 0.5, 0.75\}$, and $\eta_{\text{GD}} \in \{10^{-5}, 5 \cdot 10^{-4}, 10^{-4}, \ldots, 5 \cdot 10^{-1}, 10^{-1}\}$. In the untuned version of `Mix-IRLS`, we used the fixed values $\nu = 0.5$ and $w_{\text{th}} = 0.01$ for simulations, and $\nu = 1, w_{\text{th}} = 0.01$ for experiments.

In all algorithms, we employed the same following stopping criterion: if the estimate does not change much between subsequent iterations,

$$\frac{\sum_{k=1}^{K} \|\beta_k^{(t)} - \beta_k^{(t-1)}\|^2}{\sum_{k=1}^{K} \|\beta_k^{(t)}\|^2} < \delta^2$$

where $\delta$ is a tolerance constant, the algorithm is stopped. The tolerance $\delta$ is set to $\tilde{\delta} \equiv \min(1, \max\{0.01\sigma, 2\epsilon_{\text{mp}}\})$ in `Mix-IRLS`, `AltMin` and `EM`, and to $0.01\tilde{\delta}$ in GD.

**Additional simulation details.**  As described in the main text, the failure probability is defined as the percentage of runs with $F_{\text{latent}} > F_{\text{th}} \equiv 2\sigma$. Let us justify the choice of scaling with $\sigma$; the numerical coefficient 2 is arbitrary, and the results are insensitive to its choice. In well-defined standard linear regression (namely, with sample size above the information limit), the OLS error goes like $\sigma\sqrt{d/n}$. However, scaling $F_{\text{th}}$ with $\sigma\sqrt{d/n}$ would make the failure probability invariant to the sample size $n$. Since we want to see how the different methods improve with increasing sample sizes, we should set $F_{\text{th}}$ by the OLS error at the information limit $n = d$, so that it scales as $\sigma$. In MLR, our error measure (6) goes like $\sigma\sqrt{d/n_k}$, where $n_k$ is the number of samples that belong to the component $k$ with largest error. At the information limit $n_k = d/\min(p) \equiv d/p_K$, we get the scaling $\sigma \cdot \sqrt{p_K/p_k}$. This quantity is upper bounded by $\sigma$. Moreover, for all mixture proportions considered in our paper, this quantity is lower bounded by $\sigma/3$. Hence, also in MLR, the error threshold $F_{\text{th}}$ scales with $\sigma$.

We remark that empirically, this definition of $F_{\text{th}}$ is consistent with the critical sample sizes. At a critical sample size, the median error undergoes a phase transition: e.g., in Figure H.4, the critical sample size is $n \approx 3500$ for `Mix-IRLS` and $n \approx 8000$ for `EM` and GD. In the various figures, the failure probability at the critical sample size is roughly $50\%$, implying the consistency of the definition of $F_{\text{th}}$; compare, for example, the two panels in Figure H.5 or in Figure H.6.

**Additional experimental details.**  In the real-data experiments (Section 5), we ignore nominal fields and consider only numeric and ordinal ones. Nominal fields with two categories are considered ordinal. Table G.1 details the number of samples and the dimension in each dataset. The data is centralized and normalized as follows: $x_i \leftarrow (x_i - \bar{x}_i)/\|x_i - \bar{x}_i\|$ and $y \leftarrow (y - \bar{y})/\|y - \bar{y}\|$, where $\bar{u}$ represents the mean of a vector $u$. A bias (intercept) term was added in the medical insurance cost, red wine quality and WHO life expectancy datasets. In the CO2 emission by vehicles and fish market datasets, such a term makes no physical sense.

Finally, we remark that several authors proposed tensor-based initialization methods for MLR [6, 44, 53, 56]. In this work, we focus on random initialization, as it is more frequently used in real-data applications. For completeness, we also run the initialization procedure proposed in [56] using the code generously provided to us by the authors. However, in the relatively low sampling setting explored in this paper, this initialization did not seem to be more accurate than a random one.

## H  Additional Simulation Results

**Different sample sizes** $n$**.**  In Figure 2, we showed the failure probability of the algorithms as a function of the sample size $n$. In Figure H.3, we report the median runtime (in seconds) of the algorithms. Except for GD, the different algorithms have comparable runtimes. Figure H.4 shows the median error of the algorithms.

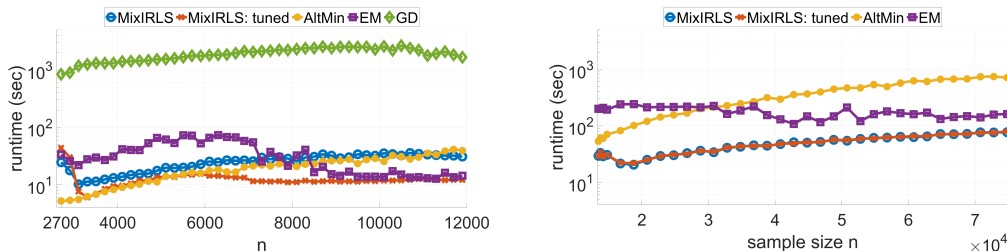

Figure H.3: Median runtime comparison in the setting of Figure 2.

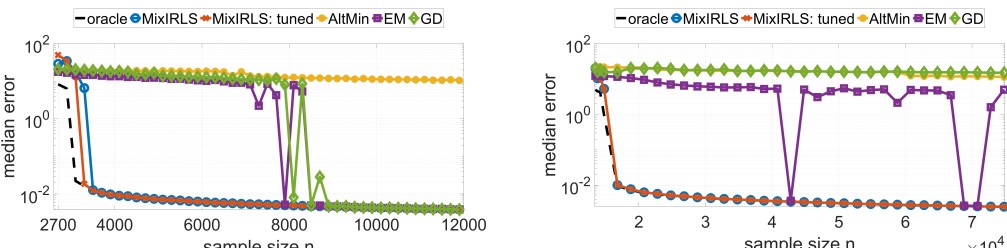

Figure H.4: Same setting as in Figure 2, but with median errors instead of failure percentage.

**Mixtures with $K = 4, 5$.** The results for mixtures with $K = 4$ and $K = 5$, including median error, failure probability and runtimes, are depicted in Figures H.5 to H.7. Specifically, Figures H.6 and H.7(right) show the results for a moderately imbalanced mixture. In all these settings, Mix-IRLS significantly outperforms the compared methods.

**Grid of sample size $n$ and dimension $d$.** Figure H.8 shows the performance of the algorithms on a 2D grid covering a broad range of values for both the sample size $n$ and the dimension $d$. As in previous simulations, Mix-IRLS recovers the linear models very close to the information limit, with negligible differences from the oracle's performance. In contrast, the other methods need much larger samples sizes to succeed in the recovery. In this simulation, we additionally included our implementation of the ILTS algorithm [45]. In contrast to the other algorithms, ILTS gets as input estimates for the mixture proportions $p_k$. In Figure H.8, we show the performance of an ILTS:latent version, which is supplied with the exact mixture proportions (this information is inaccessible to the other algorithms except for the oracle).

**Extended robustness analysis.** We extend the robustness analysis of Figure 3 in several aspects. First, Figure 3 showed only the median error of the algorithms. In Figure H.9, we show the

Table G.1: The number of samples $n$ and the dimension $d$ in each of the datasets, ignoring NaN samples and nominal fields.

| dataset name | number of samples $n$ | dimension $d$ |
|---|---|---|
| CO2 emission by vehicles[1] | 7384 | 6 |
| medical insurance cost[2] | 1338 | 7 |
| red wine quality[3] | 1599 | 12 |
| WHO life expectancy[4] | 1649 | 21 |
| fish market[5] | 159 | 5 |

[1]`kaggle.com/datasets/debajyotipodder/co2-emission-by-vehicles`

[2]`kaggle.com/datasets/mirichoi0218/insurance`

[3]`kaggle.com/datasets/uciml/red-wine-quality-cortez-et-al-2009`

[4]`kaggle.com/datasets/kumarajarshi/life-expectancy-who`

[5]`kaggle.com/datasets/aungpyaeap/fish-market`

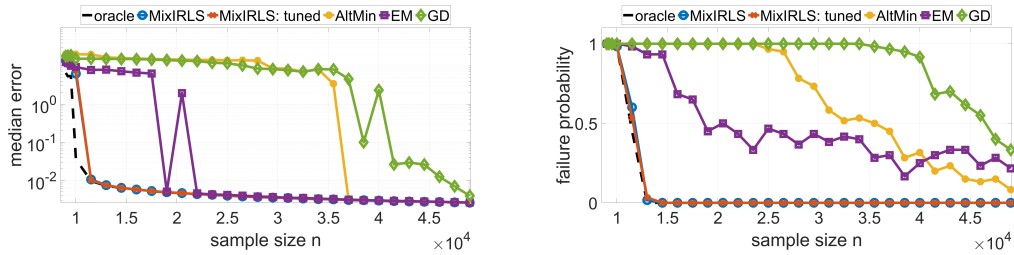

Figure H.5: Same setting as in Figure 2, but with $K = 4$ and $p = (0.67, 0.2, 0.1, 0.03)$.

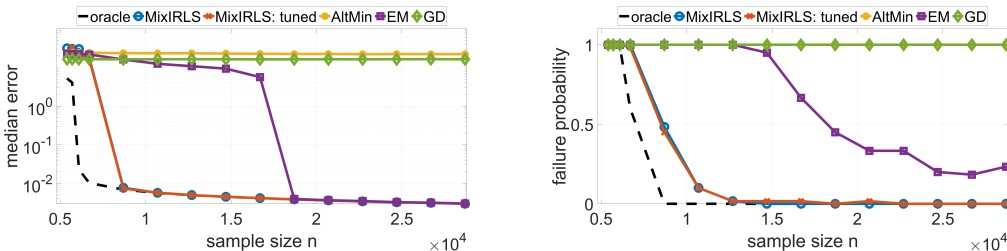

Figure H.6: Same setting as in Figure 2, but with $K = 5$ and $p = (0.4, 0.3, 0.15, 0.1, 0.05)$.

corresponding failure probability. Second, Figure 3 showed only the robustness to outliers and to overparameterization. In Figure H.10, we show the stability of the algorithms to varying noise levels. Third, we simulate different noise levels for the components. Specifically, Figure H.11 depicts the results for the setting $\sigma_1 = 10^{-1}$, $\sigma_2 = 10^{-2}$, and $\sigma_3$ ranging from $10^{-6}$ to 1. The overall picture is qualitatively similar to the uniform noise level case: most methods match oracle's performance in their median error, but only `Mix-IRLS` does it in all runs. As the results are similar to Figure H.10, for conciseness we only show the median errors and not the failure probabilities. Fourth, in Figure H.12, we examine the special case of overparameterization where the true underlying number of component is one, $K^* = 1$. In contrast to the multi-component case ($K^* > 1$), here all algorithms have relatively small median errors; however, only `Mix-IRLS` and `GD` match the oracle's performance.

**Different separation levels $\|\Delta\|$.** In this subsection we study the performance of the algorithms on mixtures with different separation levels. The separation level is defined as the distance between the components, $\|\beta_i - \beta_j\|$ for $i \neq j$; in case of more than two components ($K > 2$) we consider the average distance. A-priori, we expect to see a small error in two extreme cases: when the separation is very small, because the mixture is close to being degenerate (with the smallest error at $\Delta = 0$, corresponding to a single linear model); and when the separation is large, as the component identification is easier. In between, the error can be larger. We conducted a simulation with several separation levels, and the results are depicted in Figure H.13. We observed an interesting phenomenon: while both `Mix-IRLS` and `EM` follow the expected behavior, their peak error lies in different values of $\|\Delta\|$. At small separation levels, `EM` has a smaller error than `Mix-IRLS`, and at moderate levels it is `Mix-IRLS` which has a smaller error. However, as shown in the figure, in all separation levels `Mix-IRLS` has a better intersection score (as defined in (7)). The reason is that at smaller separation levels, `EM` finds an almost perfectly balanced mixture, which is far from the true mixture proportions $p = (0.7, 0.2, 0.1)$. In contrast, `Mix-IRLS` better clusters the samples into components and approximates the mixture proportions well even when the error in $\beta$ is relatively large.

**Perfectly balanced mixture.** We compare the performance of the algorithms on a perfectly balanced mixture, with $p = (1/3, 1/3, 1/3)$. The results, depicted in Figures H.14 to H.17, show that in this setting `Mix-IRLS` loses its advantage and performs comparably to other methods in terms of sample complexity, but holds its lead in terms of robustness to outliers and to overparameterization.

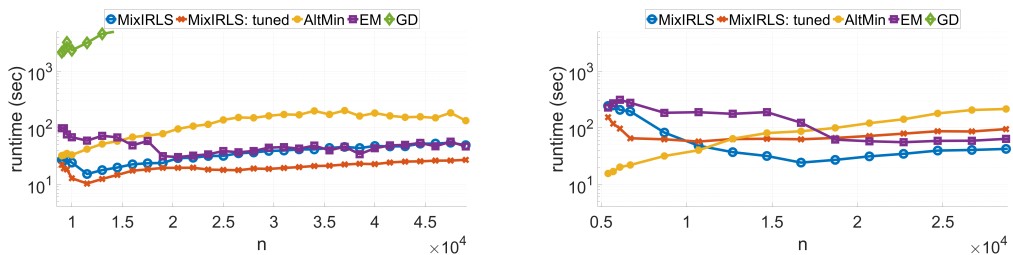

Figure H.7: Median runtimes comparison in the setting of Figure H.5 (left panel) and of Figure H.6 (right panel).

**Non-adversarial robust regression.** Intuitively, it might not be clear how `Mix-IRLS` succeeds at recovering imbalanced mixtures where the dominant component consists of less than 50% of the data. Indeed, it is well-known that the breakdown point of regression in the presence of outliers is at 50%. However, this holds in the most general setting, which allows for adversarial outliers. In the non-adversarial setting, it is possible to perform regression even with 70% or 80% outliers. Figure H.18 demonstrates this claim. In this simulation, we generated a single-component mixture ($K = 1$), with $d = 10^3$, $n = 10^4$, $\sigma = 10^{-2}$, and different fractions of corrupted samples given to the algorithm. The algorithm, which performs IRLS-based robust regression, has a breakdown point at as large as 80% outliers.

# I  Additional Experimental Results

As discussed in the main text, `Mix-IRLS` finds $K = 2$ components in the tone perception experiment given its default parameters (detailed in Appendix G). With the value of the sensitivity parameter $w_{\text{th}}$ set to $0.1$ instead of $0.01$, `Mix-IRLS` finds $K = 3$ components, as depicted in Figure I.19(right). With $w_{\text{th}} = 0.5$, `Mix-IRLS` already finds $K = 4$ components. Figure I.19(left) shows that equipped with a prior knowledge of $f = 3\%$ corruptions, `Mix-IRLS` identifies reasonable outliers. We note that this value of $f$ was chosen arbitrarily, and we do not know the true number of outliers in Cohen's data.

In Figure 4(right), we showed the median estimation errors of the algorithms for the medical insurance cost and the wine quality datasets. Figures I.20 and I.21 show the results for the WHO life expectancy, the fish market and the wine quality datasets. Most interesting is the case of the wine quality dataset with $K = 7$ components (Figure I.21). Unlike the other datasets, the response $y$ in this dataset is discrete, taking the values from 3 to 8. Hence, with $K \geq 6$, it is possible to perfectly fit an MLR model up to machine precision error. Notably, `Mix-IRLS:tuned` is the only algorithm that achieves this error with $K = 7$ in at least half of the realizations.

Finally, Table I.2 shows the minimal estimation error across different random initializations for all four datasets. Interestingly, even though `AltMin` performs consistently worse than `EM` in terms of the median error, it outperforms it in terms of the minimal error. Moreover, in the special case of $K = 2$, `AltMin` achieves the lowest minimal error in all four datasets. However, in general, the tuned variant of `Mix-IRLS` outperforms the compared methods, including `AltMin`, also in terms of the minimal error.

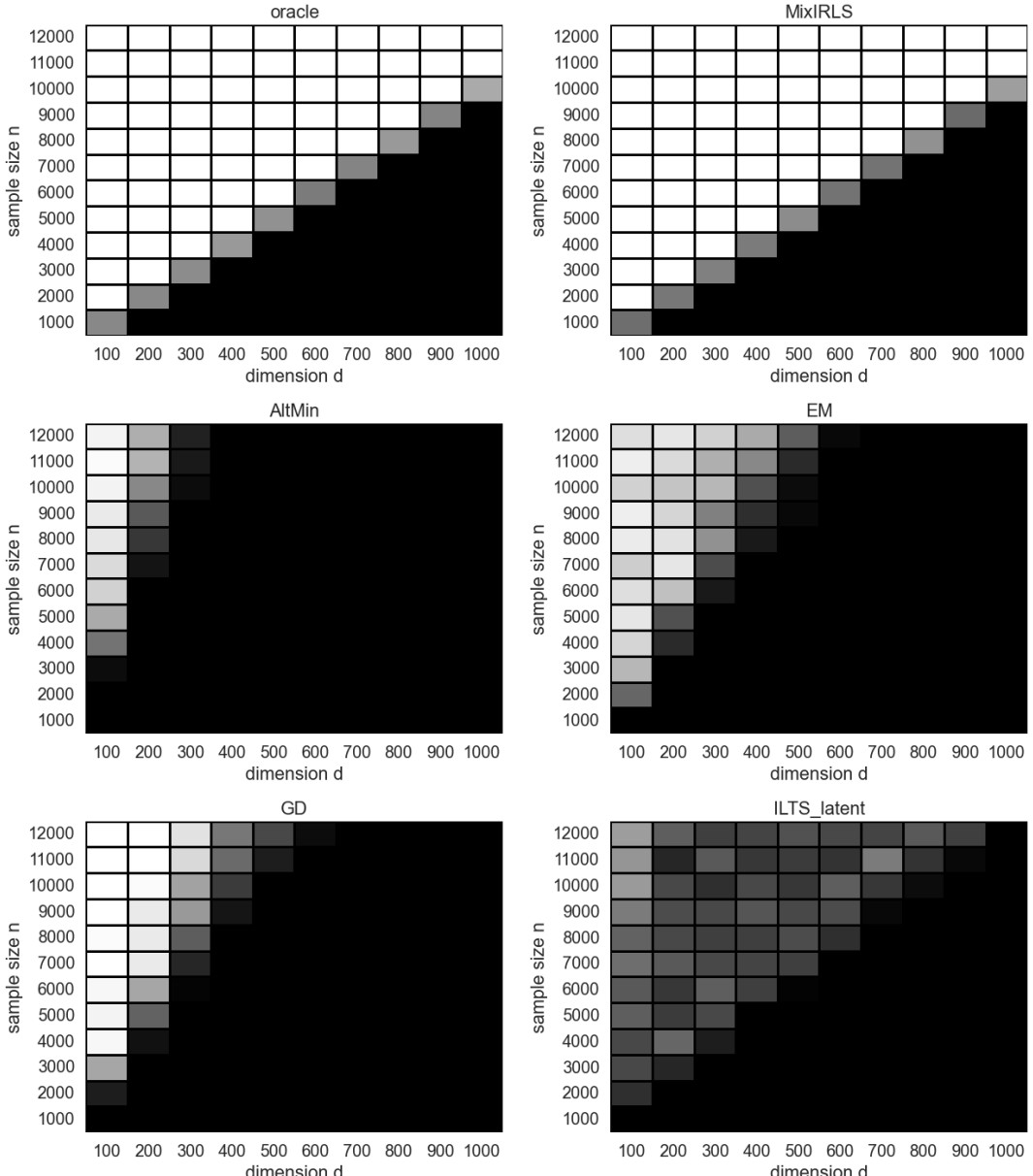

Figure H.8: The success percentage of various MLR algorithms, compared to an oracle, as a function of the dimension $d$ and the sample size $n$, with $K = 3$, $p = (0.7, 0.2, 0.1)$ and no noise $\sigma = 0$. White cell means 100% success. Comparison of the top two panels show that `Mix-IRLS` recovery is nearly as good as that of the oracle, whereas other methods require many more samples to succeed. The result for `Mix-IRLS:tuned` is very similar to that of `Mix-IRLS`, and is thus omitted.

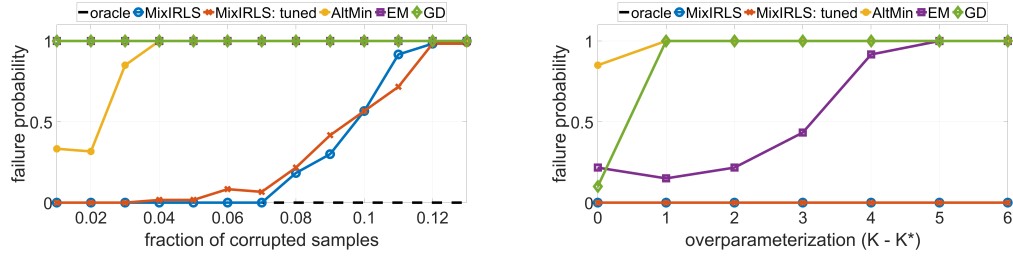

Figure H.9: Same setting as in Figure 3, but with y-axis showing the failure percentage instead of the median error.

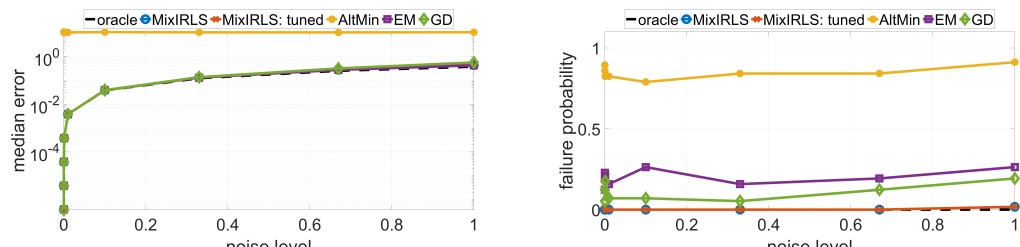

Figure H.10: Comparison of the stability of several MLR algorithms to additive Gaussian noise of mean 0 and varying standard deviation $\sigma \in [0, 1]$, for the same values of $d, K$ and $p$ as in Figure 3.

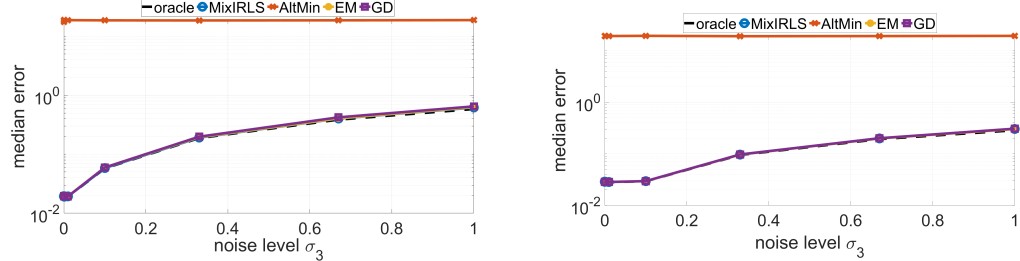

Figure H.11: Left panel: same setting as in Figure H.10, but with different noise levels for the different components: $\sigma_1 = 10^{-1}, \sigma_2 = 10^{-2}$, and x-axis is $\sigma_3$. Right panel: same, but with a perfectly balanced mixture.

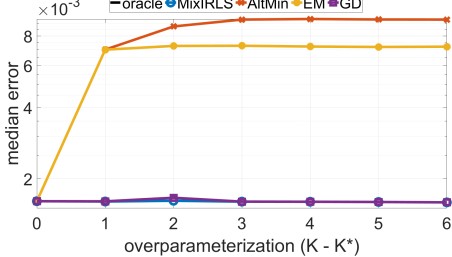

Figure H.12: Comparison of the robustness of several MLR algorithms to overparameterization for the same setting as in Figure 3, but with a single true component, $K^* = 1$.

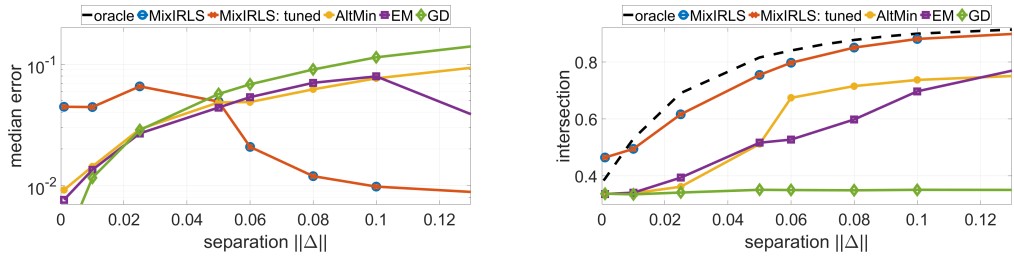

Figure H.13: Same setting as in Figure 2, but with different separation levels $\|\Delta\|$. The right panel depicts the intersection score, defined in (7).

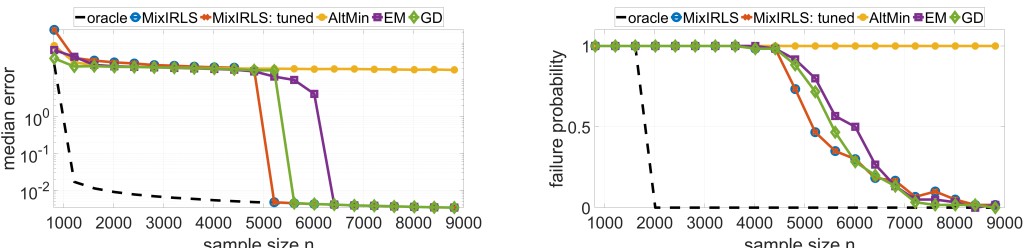

Figure H.14: Same setting as in Figure 2, but with $p = (1/3, 1/3, 1/3)$.

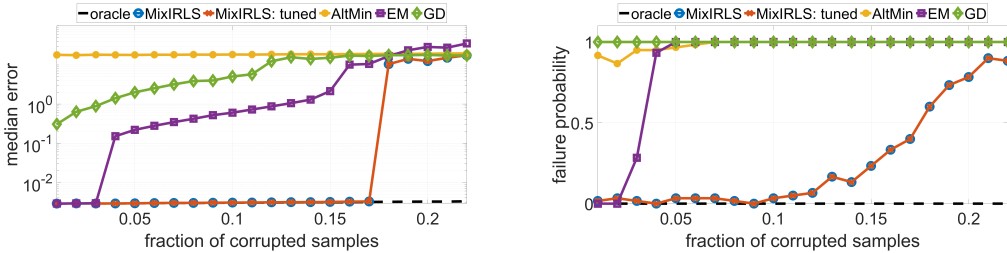

Figure H.15: Same setting as in Figure 3(left), but with $p = (1/3, 1/3, 1/3)$.

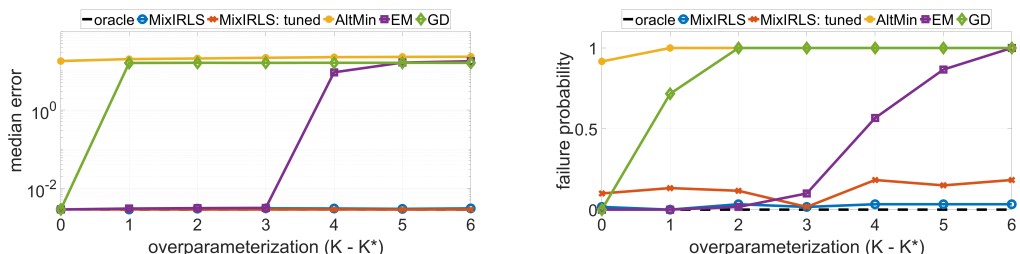

Figure H.16: Same setting as in Figure 3(right), but with $p = (1/3, 1/3, 1/3)$.

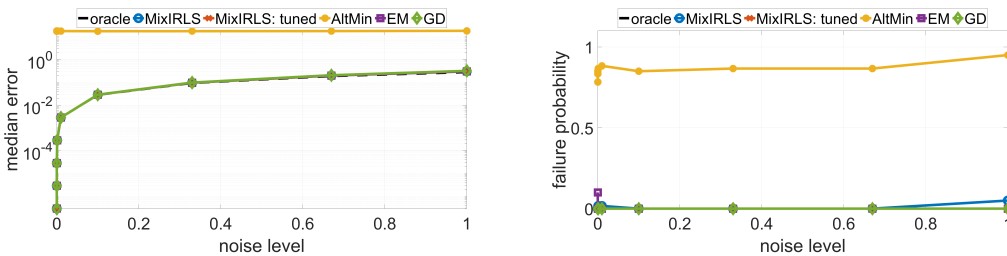

Figure H.17: Same setting as in Figure H.10, but with $p = (1/3, 1/3, 1/3)$.

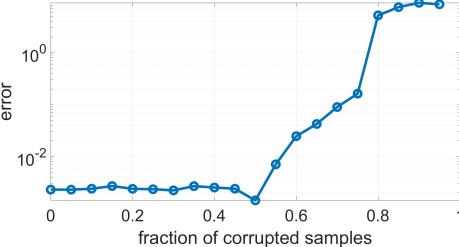

Figure H.18: Illustration of the breakdown point of an IRLS-based robust regression algorithm in a single-component setting.

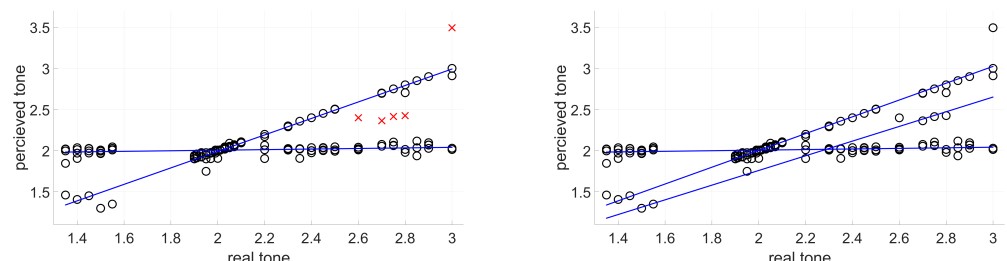

Figure I.19: Same as Figure 4(left), but with a given corruption fraction of $0.03$ (left panel), and with an increased value of the parameter $w_{\text{th}}$ (right panel). Marked with red X are outliers identified by `Mix-IRLS`.

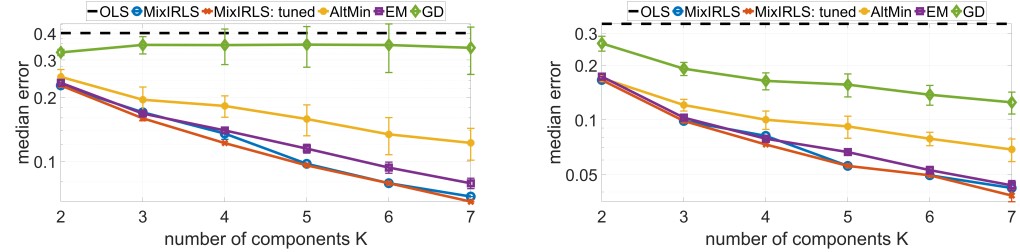

Figure I.20: Same as in Figure 4(right), but on the WHO life expectancy and the fish market datasets.

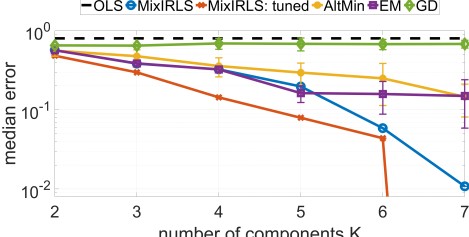

Figure I.21: Same as in Figure 4(right), but on the red wine quality dataset. While not shown in the figure, in the right panel the median error of `Mix-IRLS` with $K = 7$ components is $6 \cdot 10^{-15}$ (machine precision).

Table I.2: The minimal error, as defined in (8), achieved by several MLR algorithms, across 50 realizations of the experiment, each with a different random initialization. The corresponding median errors are depicted in Figures 4, I.20 and I.21.

| Dataset | Algorithm | $K = 2$ | $K = 3$ | $K = 4$ | $K = 5$ | $K = 6$ | $K = 7$ |
|---------|-----------|---------|---------|---------|---------|---------|---------|
| medical | Mix-IRLS | 0.1594 | 0.1086 | **0.0725** | 0.0579 | 0.0450 | **0.0428** |
|         | Mix-IRLS:tuned | 0.1594 | **0.0905** | **0.0725** | **0.0538** | **0.0435** | 0.0430 |
|         | AltMin | **0.1591** | 0.0950 | 0.0900 | 0.0653 | 0.0638 | 0.0628 |
|         | EM | 0.1598 | 0.1230 | 0.0817 | 0.0699 | 0.0589 | 0.0438 |
|         | GD | 0.2676 | 0.2567 | 0.3063 | 0.2767 | 0.2458 | 0.2920 |
| wine | Mix-IRLS | 0.4827 | 0.3836 | 0.2856 | 0.0802 | 0.0587 | 0.0108 |
|         | Mix-IRLS:tuned | 0.4827 | **0.2974** | **0.1437** | **0.0764** | 0.0438 | **0.0000** |
|         | AltMin | **0.4747** | **0.2974** | **0.1437** | 0.0776 | **0.0311** | **0.0000** |
|         | EM | 0.5593 | 0.3490 | 0.1857 | 0.1100 | 0.0485 | 0.0233 |
|         | GD | 0.5852 | 0.5043 | 0.4592 | 0.4048 | 0.3974 | 0.3391 |
| WHO | Mix-IRLS | 0.2276 | 0.1604 | **0.1201** | 0.0973 | 0.0789 | 0.0663 |
|         | Mix-IRLS:tuned | 0.2276 | **0.1517** | 0.1213 | **0.0928** | 0.0789 | **0.0646** |
|         | AltMin | **0.2246** | 0.1610 | 0.1272 | 0.1042 | 0.0974 | 0.0830 |
|         | EM | 0.2315 | 0.1604 | 0.1228 | 0.0984 | **0.0776** | 0.0653 |
|         | GD | 0.2990 | 0.2682 | 0.2281 | 0.2400 | 0.2008 | 0.1946 |
| fish | Mix-IRLS | 0.1656 | **0.0985** | 0.0808 | **0.0557** | **0.0452** | **0.0318** |
|         | Mix-IRLS:tuned | 0.1656 | **0.0985** | **0.0731** | **0.0557** | **0.0452** | 0.0327 |
|         | AltMin | **0.1637** | **0.0985** | 0.0784 | 0.0703 | 0.0551 | 0.0504 |
|         | EM | 0.1729 | 0.1027 | 0.0771 | 0.0596 | 0.0473 | 0.0362 |
|         | GD | 0.1814 | 0.1511 | 0.1188 | 0.1221 | 0.0974 | 0.0887 |