# OpenReview forum: "Imbalanced Mixed Linear Regression"
_NeurIPS.cc/2023/Conference — NeurIPS 2023 poster_

### Official Review · Reviewer_16iH · 2023-07-01

**Soundness:** 2 fair
**Presentation:** 3 good
**Contribution:** 3 good
**Rating:** 5
**Confidence:** 4

**Summary:**

The paper proposes a novel method for estimating parameters in a mixture of linear regression (MLR) model. The authors argue that the maximum likelihood estimator (MLE) does not perform well under extreme imbalanced cases and propose a sequential estimation approach using robust regression methods. The paper presents empirical results to demonstrate the effectiveness of the proposed method.

**Strengths:**

- The proposed method offers an interesting alternative to the MLE in MLR.

- The method is intuitive, and the paper is well-organized.

**Weaknesses:**

- Notations and Terminology: There are instances where the notations or terminologies used in the paper are not well-defined. For example:
    - The term "significantly imbalanced" needs clarification.
    - Line 98: What is the definition of $n_{\text{info}}$?
    - Line 190: How is median error defined? Is it the median of $F_{\text{latent}}$ over 50 repetitions? If so, how is $F_{\text{latent}}$ defined for the overspecified case?

- Method: While the intuition behind the proposed method is clear, there are several details lacking motivation or explanation. For instance:

  - What is the intuition behind equation (2b) and the weight in Phase II? How do they differ from directly running a median regression using IRLS?

  - What happens if no refinement in equation (5) is used? How does the estimator perform?

  - The method seems to rely heavily on hyperparameters such as $\rho$, $w_{\text{th}}$, $\eta$, and $T_1$. However, in Section 4, Line 73, it is claimed that "Mix-IRLS" performs well with a fixed set of values for the tuning parameters. It would be helpful to understand why the method is robust to the choice of hyperparameters.

- Experiments: More detailed descriptions of the experiments should be provided to make the results more convincing.

    - What is the definition of "median error"? How are the initial values chosen? What does "highly imbalanced proportions" mean in Line 83?
    - It is well-known that fitting mixtures becomes harder when the components are closer to each other. How similar are the different components in the experiments? What happends if the degree of similarity increases? Are these methods stil perform the same?
    - As mentioned previously, the lack of a clear definition for "median error" makes the results difficult to interpret.
    - Instead of reporting the extreme failure probabilities, it would be better to report the mean squared error (MSE) of the regression estimator. Additionally, the performance of different estimators could be compared using this metric.

- Theory:

    - Providing a definition of "significantly imbalanced" would be helpful for understanding the paper. In the abstract, it is mentioned that the method works for imbalanced cases with a "significantly different" number of samples. However, in Remark 6.2, it is stated that Theorem 6.1 holds only for a sufficiently imbalanced mixture, and $p_2<1/2$ does not necessarily imply significant imbalance.

    - While theoretical results on the gap between the output of the proposed method and the true value under the population case is interesting, they are not useful in practice where we have a finite number of samples. It would be valuable to demonstrate 1) the consistency of the estimator and 2) the rate of convergence of the proposed method.

    - For cases where the number of components $K$ is unknown, the theoretical results in Appendix C only state that the algorithm stops under the given criteria, but they do not describe any properties of the estimator.

**Questions:**

- The mixture model in (1), are the explanatory variables random or fixed? It looks like they are random based on the theory. Should be specified when the model is defined.

- The definition of the mixing proportion is weird in the sense that it depends on the sample size $n$. Should it be $p_k=P(c_i^*=k)$ instead?

- The statement that "Then the information limit .... is $n_{\text{info}}", can you please explain? Similarly for the statment that "A suitable value for $\rho$ depends on the ratio ...".

- $P(\epsilon_t=t)=P(\epsilon_t=-t)$ does not mean anything for continuous random variables, should use CDF or pdf instead.

**Limitations:**

Some limitations and possible future research directions are briefly mentioned in the paper.

---

> ### Author Rebuttal · Authors · 2023-08-04
>
> We thank the reviewer for their detailed and constructive feedback. Below we address each point separately. We will carefully revise our manuscript according to the points raised by the reviewer.
>
> **Weaknesses**:
> * Notation and Terminology:
>   - By "significantly imbalanced" we mean that the mixture is far from being balanced, e.g. $p=(0.7,0.2,0.1)$. We will reword this term in the revised manuscript to make our point clearer.
>   - The definition of $n_\text{info}$ is the minimal sample size $n$ that makes the problem identifiable. Kindly see item 3 in our global response for more details.
>   - Yes, the median error is the median of $F_\text{latent}$ over 50 repetitions. As the reviewer implies, the error is defined differently in the overspecified case, with the true number of components $K^*$ replacing $K$ in Eq. (6); please see Lines 218-219. Namely, in the overspecified case, only the best $K^*$ (out of $K$) components of the estimate are compared to the ground truth.
> * Method:
>   - We thank the reviewer for bringing this unclarity to our attention. Eq. (2b) employs a standard Cauchy weighting, which was found empirically to suit our needs. However, other standard schemes could be used as well. The scheme is similar in phase II but involves two differences tailored to our needs. First, since the weights are calculated for all components, they are subsequently normalized over the components such that $\sum_{k=1}^K w_{i,k}=1$. Due to this normalization, we do not need the weights to lie between $0$ and $1$ as in standard weighting schemes; so instead of $1/(r^2+1)$ (Cauchy) we use $1/(r^2+\epsilon_\text{mp})$ (Eq. (B.1b)). This modification encourages the dominance of one component over the others. We will include this discussion in the revised version of our manuscript.
>   - In practice, the effect on the refinement in Eq. (5) is negligible. However, conceptually, it makes sense to use only the largest-weighted samples - namely, the samples that belong to the estimated component with high confidence - to re-estimate the component. We will clarify this point in the revised manuscript.
>   - Thanks for the question. Let us focus, for example, on the oversampling parameter $\rho$. To estimate a component, we must use at least $d$ samples, otherwise the system is underdetermined. In our algorithm, we use (the largest-weighted) $\rho\cdot d$ samples, $\rho\geq 1$; see Eq. (4). If the sample size is large enough, then $\rho$ has little effect: we can use many samples to estimate a component ($\rho\gg 1$), and still have enough samples for the next components. However, in extreme cases where the sample size is close to the information limit, $\rho$ should be close to 1, so that we use the minimal number of samples for each component. This idea is demonstrated in Fig. 2(right) in the PDF attached to our global response. It is shown that MixIRLS starts to perform worse when $\rho>n/n_\text{info}$, as expected. Due to the space limit, we cannot address the other parameters here, but we will do it in the revised manuscript (or during the discussion period, if requested).
> * Experiments:
>   - In the simulations section, the error is defined as $F_\text{latent}$ (Eq. (6)), and in the experiments section it is $F_\text{real}$ (Eq. (7)). In both cases, the median error is taken over 50 repetitions, each with a different random initialization (but the same one for all algorithms). The initialization is drawn element-wise from the normal distribution, $\beta_{k,i}\sim N(0,1)$. "Highly imbalanced proportions" means that the mixture is far from being balanced, e.g. $p=(0.7,0.2,0.1)$, whereas $p=(0.4,0.3,0.3)$ is only a "moderately imbalanced mixture".
>   - We thank the reviewer for bringing up this point. In our simulations, the components are well-separated. We agree that examining different separation levels would be very interesting. Kindly see item 2 in our global response.
>   - We hope the definition of "median error" is now clear.
>   - In addition to the failure probability, the MSE of all estimators in all simulations is also reported. However, due to space limits, most of these results appear in Appendix F.
> * Theory:
>   - We apologize for this misunderstanding, which will be clarified in the revised version. Empirically, MixIRLS works best for mixtures that are far from being balanced, e.g. $p=(0.7,0.2,0.1)$. However, it works well also for moderately and even perfectly balanced mixtures. Our theory supports these findings: In the general case, our theorem provides a guarantee for MixIRLS only for a sufficiently imbalanced mixture. In the special noise-free case, our theorem holds for an almost perfectly balanced mixture, $p_2<1/2$.
>   - Unfortunately, we do not currently have a finite-sample analysis for our algorithm, but we plan to accomplish it in future research. Kindly see item 4 in our global response for more details on the challenges in finite-sample analysis
>   - Thanks for spotting this. We forgot to explicitly write it in Proposition C.1, but our proof shows that the estimator has the same error in the case of unknown $K$ as in the known $K$ case (in fact, the estimators are identical). We will reword Appendix C accordingly.
>
> **Questions:**
>   - The explanatory variables in the MLR model (1) can be either fixed or random. In our theory, we assume random explanatory variables to simplify the analysis.
>   - Since we do not necessarily assume a random model, the label vector $c^*$ can be fixed. Hence, we cannot define the mixture proportions $p_k$ by $P(c_i^*=k)$. Instead, $p_k$ is defined as the empirical proportion in the observed data, and is thus dependent on the sample size $n$.
>   - For an explanation of $n_\text{info}$, kindly see item 3 in our global response. For an explanation of the relation between $\rho$ and $n_\text{info}$, please see the last item under the "Method" title (above in this response).
>   - Thanks, we will replace the probability notation in Eq. (9) with CDF.

---

> > ### Comment · Reviewer_16iH · 2023-08-16
> >
> > Thanks for your rebuttal. While I like the new approach to learn mixture of regression model, I still think it would make the paper stronger if its advantageous over the conventional MLE can be characterized more precisely. Therefore, I decide to keep my current score.

---

### Official Review · Reviewer_Yapb · 2023-07-04

**Soundness:** 2 fair
**Presentation:** 2 fair
**Contribution:** 2 fair
**Rating:** 6
**Confidence:** 3

**Summary:**

This paper introduces a novel algorithm for mixed linear regression suitable for balanced and imbalanced mixture settings. It proves the proposed method holds for a sufficiently imbalanced mixture rather than a balanced one through a recovery guarantee on the theoretical front. Empirical results on synthetic data and real-world datasets highlight the superior performance of the proposed algorithm compared to conventional methods. The key contributions of this paper can be summarized as follows:
+ The paper presents an efficient and straightforward approach to addressing imbalanced mixed linear regression.
+ It proposes to find the dominant model, remove its associated samples from the observation set, and repeat the process to find the next dominant model.
+ It demonstrates the effectiveness through empirical results and theoretical analysis.

**Strengths:**

+ The paper is well-organized and easy to follow.
+ The proposed method is efficient and simple. The central idea of simultaneously recovering the models while partitioning the samples into three classes based on their fit to the estimated model is indeed intriguing and adds value to the overall approach.
+ This approach seems promising for addressing mixed linear regression in both balanced and imbalanced mixture settings, as supported by the empirical results and theoretical analysis presented in the paper.

**Weaknesses:**

+ My primary concern revolves around the motivation of this paper. While the title focuses on "imbalanced mixtures," the abstract and introduction cover a wide range of settings, such as small sample sizes, the presence of outliers, and an unknown number of models. While it is commendable that the authors aim to demonstrate the robustness of the proposed method, it would be preferable to emphasize one main topic to maintain clarity and focus.

+ Figure 1 visually illustrates the central idea of the proposed method, but further analysis or additional figure tools are required to explain why this method is particularly suitable for imbalanced settings. Providing in-depth insights into the algorithm's performance in such scenarios would enhance the paper's credibility.

+ In Figure 2, a comparison with conventional mixed linear regression methods is presented, but the selected methods mainly date back to before 2016, which might not reflect the state-of-the-art results. Including comparisons with more recent methods would bolster the paper's claims about the effectiveness of the proposed approach. Additionally, line charts, while showing the experiment trends, may not be sufficient to fully convey the experimental findings. Further detailed analysis and discussion of the results are necessary.

**Questions:**

Please refer to the "Weaknesses" part.

**Limitations:**

The authors should clarify the potential negative societal impact of their work.

---

> ### Author Rebuttal · Authors · 2023-08-06
>
> We thank the reviewer for the helpful comments.
>
> * We thank the reviewer for bringing this issue to our attention. Our goal was to emphasize the advantages of our method, but we understand that the reader might miss our main message. We will revise our abstract and introduction and focus on our method's success in the imbalanced mixture setting.
> * Following the reviewer's comment, we added two new figures to our manuscript. The first one is Fig. 1 in the PDF attached to our global response. This figure illustrates the algorithm's performance given an imbalanced mixture, specifically with the proportions $p = (0.7, 0.2, 0.1)$. Interestingly, at the first iterations, the estimate of MixIRLS is closer to the second component, whose proportion is $p_2 = 0.2$. This must be a consequence of the random initialization, which happened to be closer to the second component. It is further shown that as the iterations proceed, MixIRLS gradually sets its focus to the dominant component ($p_1 = 0.7$). The second figure included in the revised version shows the performance of our IRLS variant as a function of the number of outliers. As expected, IRLS performs better as the number of outliers decreases, corresponding to a higher imbalance in the MLR setting.
> * We compared MixIRLS to popular methods for MLR. Specifically, EM is by far the most popular one, and is considered the state-of-the-art also to these days. More recent works either focused on theory and have no available implementation for their method, or adapted and analyzed EM for other variants of the MLR problem (e.g., sparse MLR), and are thus incomparable to our work.
> * We will further analyze and discuss our experimental results in the revised manuscript. Specifically, we will analyze and interpret our results (i) in the Kaggle datasets and (ii) in the newly added CO2 emission dataset. Kindly see item 1 in our global response for more details.

---

> > ### Comment · Reviewer_Yapb · 2023-08-18
> >
> > Thank you for the author's rebuttal, which largely addressed my concerns. Taking into account the feedback from other reviewers and the newly added CO2 experiment section, I will raise my score to 6.

---

### Official Review · Reviewer_2aVS · 2023-07-06

**Soundness:** 3 good
**Presentation:** 3 good
**Contribution:** 3 good
**Rating:** 7
**Confidence:** 4

**Summary:**

In this paper, the authors take into account the problem of recovering a mixture of $K$ unknown linear regressions with imbalanced proportions, i.e. the numbers of samples for components in the mixture are substantially different. Rather than try to recover $K$ components at the same time as in previous work, they propose an algorithm named Mix-IRLS which finds these components sequentially using robust regression. This algorithm is theoretically shown to be able to recover the components with small sample size regardless of information about the true number of components $K$. Additionally, it is also demonstrated to be robust to noise and outliers. Finally, the authors empirically polish the performance of the Mix-IRLS algorithm on imbalanced mixtures by conducting simulation studies on both synthetic and real-word datasets.

**Strengths:**

1. Originality: This is the first work to tackle the imbalance in the mixed linear regression. It provides a novel viewpoint of recovering mixture components sequentially rather than simultaneously as in the literature.

2. Quality: Results in this paper are empirically supported and some of them are theoretically grounded. The proposed method is technically sound and applicable for high-dimensional settings.

3. Clarity: The paper is well-written and easy to follow. Assumptions are discussed clearly. Proofs are presented neatly.

4. Significance: This work is important as it is the pioneer in solving the imbalanced problem of mixed linear regression.



**Weaknesses:**

1. The population setting considered in Theorem 6.1 makes this result not practical as we do not have access to an infinite number of samples in practice.

2. Theoretical guarantee for the robustness to outliers and noise of the Mix-IRLS algorithm is missing.

3. There are some claims not associated with any proofs or references. A typical example is a claim in lines 98 & 99: 'No method can recover the models given fewer samples'.


**Questions:**

1. In line 124, should the number of iterations $T_1$ vary with $k$? In particular, the number of samples of interest decreases after each round, which possibly affects the convergence of parameters.

2. In line 135, I think one of the conditions for S'k and $S_{k+1}$ not being disjoint is that the threshold $\omega_{th}$ is too high, not too low as claimed in the paper because the higher the $\omega_{th}$ is the more elements the set $S_{k+1}$ has.

3. Are there any theoretical guarantee that by following the procedures to find the true number of components $K*$, the output is the desired value?

4. What are the main challenges of showing Theorem 6.1 under the settings with the finite number of samples?

**Limitations:**

The authors already discussed the limitations of the proposed method in this work.

---

> ### Author Rebuttal · Authors · 2023-08-06
>
> We thank the reviewer for their constructive feedback.
>
> **Weaknesses**:
> 1. Unfortunately, we do not currently have a finite-sample analysis or a robustness to outliers guarantee for our algorithm. We agree that these two guarantees would greatly improve the theoretical grounding of our method, and we plan to accomplish them in future research. On the other hand, we do have a robustness to noise guarantee, as described in Section 6. Our guarantee holds for a bounded and symmetric noise.
> 2. Addressed in 1 above.
> 3. We thank the reviewer for bringing this unclear point to our attention. Kindly see item 3 in our global response. We will further look for and fix any other unjustified claim.
>
> **Questions**:
> 1. We agree with the reviewer that a more efficient scheme would lower the value of $T_1$ with $k$. However, it seems nontrivial to devise a suitable scheme provided that the mixture proportions are unknown. Furthermore, the number of iterations done in practice is almost always much less than $T_1$, since we employ an early stopping criterion. This criterion is described in Appendix E.
> 2. Thanks for spotting this. We will correct it in the revised manuscript.
> 3. Yes. Under the conditions of Theorem 6.1, MixIRLS provably finds the true number of components; please see also Remark 6.4. This claim is formally stated in Appendix C (Proposition C.1), and proved in Appendix D.
> 4. Thanks for this question. Kindly see item 4 in our global response.

---

> > ### Comment · Reviewer_2aVS · 2023-08-12
> >
> > Thanks for the rebuttal, which addressed all of my concerns. Given what we have discussed, I think the rating of 7 is reasonable, so I decide to keep my rate unchanged.

---

### Official Review · Reviewer_1Ro7 · 2023-07-06

**Soundness:** 3 good
**Presentation:** 3 good
**Contribution:** 3 good
**Rating:** 7
**Confidence:** 4

**Summary:**

In this paper the authors study the problem of mixed linear regression, where instead of a single linear regression model, the response variable comes from a finite mixture of them. The authors propose an algorithm for estimating the regression coefficients, especially in the imbalanced case where the mixing proportions can be very different from each other. The algorithm is efficient and scalable to high dimensions, does not require knowledge of the number of components, and is robust to noise and outliers. The algorithm is tested on both synthetic and real data sets, outperforming several existing methods. Theoretical aspects of the algorithm are also analyzed on the population level in a two-component model.


**Strengths:**

1. The paper presents a conceptually simple algorithm. Although the idea of sequentially fitting each component is not new, the incorporation of robust statistics leads to an efficient and scalable algorithm that outperforms several other standard algorithms for mixed linear regression on both synthetic and real data.

2. The authors show that the algorithm does not require the knowledge of the number of components, which is a very favorable property for learning mixture models, along with robustness to noise and outliers.

3. The algorithm handles sufficiently imbalanced mixtures, extending existing literature which is mostly for balanced or sufficiently balanced cases. It also gives comparable and more robust estimation even in the perfectly balanced case.

4. Although the algorithm requires several hyper parameters, the authors show that tuning of the parameters does not lead to major improvement so the algorithm is essentially tuning free.

5. The theoretical results give a justification of the algorithm on the population level, especially in the sufficiently imbalanced case.

**Weaknesses:**

1. The design of the algorithm seems to rely heavily on the imbalanced mixing proportion assumption to be able to filter out each component, yet it gives competitive or even improved performance in the perfectly balanced case. Some intuition or explanation would be helpful since the balanced case is not covered by the theoretical results.

2. The theoretical results investigate only the infinite data setting. The paper could be strengthened with a finite sample analysis.

**Questions:**

1. If I understand correctly, the noise distribution is assumed to be the same across all the components. Does the algorithm apply in the case when the noise distribution is different for each component? Or is there any major modification needed?

2. Let's say we have a two-component model with perfectly balanced weights (p1=p2=1/2). In an intuitive sense, how would the algorithm proceed in this case? I guess the initial step would end up with something like a line right in the middle of the two components, and wonder how the algorithm would correct for this.


**Limitations:**

Theoretical analysis of the algorithm could be strengthened, but the empirical success is already very compelling.

---

> ### Author Rebuttal · Authors · 2023-08-06
>
> We thank the reviewer for their insightful comments.
>
> **Weaknesses:**
> 1. We thank the reviewer for bringing this unclear point to our attention. It might not be so well-known, but in general, robust regression algorithms succeed even when the number of outliers largely exceeds the number of inliers. The well-known 'breakdown point' at 50% allows for adversarial outliers: consider, for example, a case where all the outliers follow the same linear model. Then the problem is clearly ill-posed. However, the 'outliers' in our case, namely the samples that belong to a yet undiscovered component, are not adversarial, but rather follow an MLR model. In such cases, standard high-dimensional robust regression methods can recover the underlying linear model even with 70% and 80% outliers. This explains the success of MixIRLS in balanced mixtures from an empirical perspective. From a theoretical perspective, we would like to point out that our theorem does include a guarantee for a balanced mixture, provided that the noise is small enough. In the special case of noise-free observations, our theorem holds for all mixture proportions except for the perfectly balanced one. We will elaborate on this point in the revised version of our manuscript. Additionally, we will support our claim regarding the recovery abilities of non-adversarial robust regression with simulations.
> 2. Unfortunately, we do not currently have a finite-sample analysis for our algorithm, and we plan to accomplish it in future research. Kindly see item 4 in our global response for more details on the challenges in finite-sample analysis.
>
> **Questions:**
> 1. Yes, the algorithm applies also in the case of noise distribution which is different for each component, and no modification is needed. On the empirical front: please see Fig. F.9 in Appendix F. It is shown that the algorithms (not only ours) are not much affected by the variability of the noise levels. On the theoretical front: our theorem also holds in this case; the bound in Eq. (9) would then refer to the maximal noise level. We will clarify this point in the revised manuscript.
> 2. This is an interesting question. The algorithm would succeed also in this case. The reason is that even if the mixture is perfectly balanced with $p_1=p_2=1/2$, the different sample magnitudes $||x_i||$ and noise terms $|\epsilon_i|$ make the problem asymmetric. As such, MixIRLS will gradually tend towards one of the components, and eventually recover it. Only in the population setting perfect balance is a symmetric pathology in which MixIRLS enjoys no guarantee. We will include this discussion in the revised manuscript.

---

> > ### Comment · Reviewer_1Ro7 · 2023-08-13
> >
> > Thanks for the response, which clarifies my questions. Please include a discussion on robust regression as the authors reply to the first weakness point. I'm happy to raise my score to 7.

---

### Official Review · Reviewer_LCfo · 2023-07-18

**Soundness:** 3 good
**Presentation:** 3 good
**Contribution:** 2 fair
**Rating:** 6
**Confidence:** 3

**Summary:**

This paper proposes a new method, Mix-IRLS, for mixed linear regression, in which data are drawn from a mixure of linear models whose coefficients we wish to estimate. The method is especially motivated by the failure of existing methods in the case of imbalanced mixtures (i.e., when some components are much rarer than others). After presenting the new method, the paper presents experimental results on simulated and real data, illustrating the improved robustness of Mix-IRLS over existing methods. Finally, the paper proves that, in a particular two-component, infinite-sample example, a slight variant of Mix-IRLS is able to recover both of the true regression coefficients (even without knowing the number of components a priori).

**Strengths:**

The MLR problem is clearly presented and the proposed Mix-IRLS approach is well-motivated. The simulations clearly illustrate the basic strengths of Mix-IRLS (robustness to imbalance, contamination, and unknown K) over existing methods. The application to the music perception dataset of Cohen [13] is compelling and well-explained.

**Weaknesses:**

1) The introduction doesn't mention any applications motivating the MLR problem. The only real-world example is the music perception dataset of Cohen [13] in Section 5. I would find the paper more compelling if it were clear to the reader why this problem is useful to study. It would be especially compelling if there were real-world examples where class imbalance was a limiting issue for existing methods (the music example appears fairly balanced, although it is hard to tell from the plot).

2) Some of the assumptions in Theorem 6.1 seem arbitrary and unnecessary. Specifically, the assumption X ~ N(0, I_d) is very strong, and the resulting in the theorem doesn't illustrate how the distribution of X impacts the recovery guarantee. Based on existing theory for linear regression, I would expect a much milder assumption, such as a lower bound on the smallest eigenvalue of the covariance matrix of X, to suffice. Also, as far as I can tell, excluding X_i with ||X_i||^2 < R is done only to simplify the analysis and shouldn't really affect the recovery guarantee (indeed, as far as I can tell, the guarantee in Theorem 6.1 only improves as R -> infinity). These assumptions make Theorem 6.1 feel more like an "example" than a "theorem", and encourage the authors to work towards a more general formulation of Theorem 6.1.

Overall, on initial reading, I lean weakly towards rejection because the real-world significance of the work is not very clear and the theoretical results seem unpolished. But I could change my score if either of these points is improved substantially.

**Questions:**

1) In addition to each component of c^* being sufficiently large, I think identifiability also requires all of the \beta's to be distinct (consistent with the 1/||\Delta|| terms in Eq. (10)). Perhaps this should also be mentioned in the context of identifiability, around Line 94.

2) Lines 97-99, "the minimal number of observations n required to recover $\beta^*$ in the absence of noise is $n_{info} = d/min(p)$." What is the justification for this sentence, and what assumptions does it require? Is it somehow obvious? Can the authors prove it or provide a reference?

3) Lines 101-102, "Even in the simplified setting... the problem is NP-hard without further assumptions." How is the present work able to get around this? Does it make further assumptions? (If so, which ones?) Does it guarantee only approximate recovery? Or something else?

4) I was confused by the corruption added in the second simulation (corresponding to the left panel of Figure 3). It seems to me that the corrupted samples \tilde{y}_i are a special case of an MLS component, with \beta = 0 and a different noise scale. As far as I understand, Mix-IRLS does not know noise level \sigma, so it cannot distinguish which is the "real" data and which is the "corruption". So, it seems like Mix-IRLS is arbitrarily selecting one of the two possible MLS models, and is ignoring the other. Is this task well-specified?

5) What is the goal of applying Mix-IRLS to the Kaggle regression benchmark datasets (health insurance, red wine, WHO life expectancy, and fish market)? As far as I could tell, the purpose of Mix-IRLS is parameter (\beta) estimation, but it's not clear to me what parameters are being estimated in these examples, or how these results could be interpreted.

6) In the Problem Setup (Section 2), the distribution of explanatory variables Xs is not discussed. This is fine, since a precise statistical model is not needed (except in Theorem 6.1), but I feel like I am lacking some intuition for the problem because of this. Sometimes (e.g., Figure 1), their distribution seems to differ for each MLR component, whereas elsewhere (e.g., Theorem 6.1) their distribution is the same. Intuitively, the former (different distributions) setting seems potentially much easier than the latter (same distributions) setting, since the components will be more separated. Is there a particular way the authors suggest thinking about the distribution of Xs?
I also have a similar question regarding the distributions of \epsilons -- is it important that the noise distributions be similar across components?

**Limitations:**

Section 7 discusses some limitations of their theoretical results, but perhaps this could be more extensive (see point 2 under "Weaknesses" above).

---

> ### Author Rebuttal · Authors · 2023-08-07
>
> We thank the reviewer for their thoughtful comments.
>
> **Weaknesses:**
> 1. Thanks for bringing this up. We performed an additional experiment based on a Kaggle CO2 emission from vehicles dataset to address the reviewer's concern. This dataset approximately follows an MLR model, where the samples can be clustered into components according to their fuel type. We compared the performance of the MLR methods given the original mixture, which is imbalanced, and given a subsampled set of observations such that the resulting mixture is balanced. The results imply that the imbalance is, to some extent, a limiting issue for existing methods but not for MixIRLS. Kindly see item 1 in our global response for more details on this experiment. We will include these results and discuss them in the revised manuscript.
> 2. We agree with the reviewer that generalizing Theorem 6.1 would greatly improve our work. In fact, since we assume a population setting, we can extend our theorem to the case $X \sim N(\mu, \Sigma)$ without modifying our proof by removing the mean and whitening the samples, $x_i \to \Sigma^{-1/2} (x_i - \mu)$. We thank the reviewer for pointing us in this direction. As a side note, we would like to remark that most works that derived theoretical guarantees for MLR in the finite-sample setting - even for extensively studied methods such as EM - did have the strong assumption $X \sim N(0, I_d)$. These include [2, 21, 33, 34, 35, 36, 52, 53, 56] and other works.
> The parameter $R$ is indeed an artifact of our proof. It helps us deal with challenges that are unique to the theoretical analysis of our (sequential) approach; for more details on these challenges, please see item 4 in our global response.
>
> **Questions:**
> 1. The reviewer is correct in that separability is required to identify the components. However, separability is unnecessary if our goal is only estimating the $\beta$ vectors rather than the labels $c_i^*$. Consider, for example, the extreme case of zero separation. Then the mixture is degenerate and the problem is identical to standard linear regression. We will add this discussion to the revised manuscript. Having said that, our current manuscript does lack an analysis of the algorithms' performance with respect to different levels of separability. We ran a suitable set of simulations and described the result in item 2 in our global response.
> 2. Kindly see item 3 in our global response.
> 3. Yes, we make an additional assumption on the distribution of the explanatory variables: $X_i$ are independent and follow the normal distribution. To the best of our knowledge, this assumption is common to most works on MLR with theoretical guarantees. We will clarify this point in the revised manuscript.
> 4. This is an interesting question. Formally, the corrupted samples can be viewed as another MLR component with $\beta=0$. However, the signal-to-noise ratio in this case is infinitely large. To answer the reviewer's question regarding the behavior of MixIRLS, we need to split the discussion into two settings: known $K$ and unknown $K$. If $K$ is known, treating the corrupted samples as another component would significantly increase the MLR objective $F_\text{latent}$ (Eq. (6)) due to the large noise of this component. Hence, MixIRLS finds the true $K$ components, and the corrupted samples get low weights and are thus treated as outliers. If $K$ is unknown, the behavior of MixIRLS is the same in the first $K$ components; the key question is whether MixIRLS would incorrectly 'view' the corrupted samples as a $(K+1)$-th component. Since we provide the algorithms with the corruption fraction, the answer is that after finding $K$ components MixIRLS 'knows' it already used all the inliers in data, and does not look for new components. Hence, the corrupted samples will not be treated a $(K+1)$-th component.
> 5. Thanks for this question. Kindly see item 1 in our global response.
> 6. Thanks for highlighting this unclarity. The reviewer touches on an important question regarding a core MLR setting. In the so-called "model-based clustering" variant of the MLR problem, the distribution of the explanatory variables depends on the component (label), $x_i = x_i(c_i^*)$. In our work, we consider the vanilla "model-free clustering" setting, where the distribution of the explanatory variables is identical for all components. As noted by the reviewer, this setting is in general more challenging than the former. We apologize for the misleading visualization in Fig. 1, which will be fixed in the revised manuscript.
> As for the noise distribution, it needs not be similar across the components. On the empirical front, we show in Fig. F.9 that the algorithms (not only ours) are not much affected by variability in the noise levels. On the theoretical front, our theorem also holds in this case; the bound in Eq. (9) would then refer to the maximal noise level. We will clarify this point in the revised manuscript.

---

> > ### Comment · Reviewer_LCfo · 2023-08-16
> >
> > Thanks to the authors for their rebuttal, as well as for the follow-up class-wise analysis of the CO2 dataset. In my opinion, these additions clarify the motivation and strengthen the empirical part of the paper. In the theory part, I still find the assumption that $X$ is normally distributed to be quite restrictive. Also, as pointed out by other reviewers, finite-sample analysis would be more useful than population-level analysis. Based on these factors, I have raised my score from 4 to 6.

---

### Author Rebuttal · Authors · 2023-08-04

We thank the reviewers for their time and effort. We appreciate the constructive feedback, which will definitely improve the manuscript. Please find attached a one-page PDF with new figures that help answer some of the reviewers' concerns. Following the reviewers' comments, we describe below few important improvements to our manuscript.

1. **Real-world experiments** [in response to Reviewers LCf0, Yapb, 16iH]. In our Kaggle datasets, it is not known whether the data follow an MLR model. Hence, interpreting the results of MLR algorithms in such cases is a delicate issue. We address this issue in two ways:
    - In the revision, we will more clearly explain our results on these datasets. Our experiments demonstrate the following claim: MixIRLS finds MLR coefficients that fit the data significantly better than other methods. As a consequence, *given that* the data (approximately) follow an MLR model for some $K$, our method will often find the mixture coefficients where other methods fail.
    - We added a new experiment on the CO2 emission by vehicles dataset (also from Kaggle) to complement our empirical study. This dataset approximately follows an MLR model, with the fuel type as the label vector $c^*$ (regular/premium gasoline, diesel or ethanol), such that $K=4$ and $p\approx (0.49,0.43,0.05,0.03)$. As the labels in this dataset are known, we can compare how well each method clusters the data into components. We do so by comparing the maximal intersection over all permutations over 4 items, between the labels found by each method and the true labels: $\max_{\sigma\in [K]!}|\\{i\in [n]:\sigma(c_i)=c_i^*\\}|/n$. Table 1 in the attached PDF demonstrates the favorable performance of MixIRLS on this task: almost 6% higher than the second-best method.
In addition, following Reviewer LCfo's first comment, we further checked whether the success of MixIRLS has to do with the imbalance of the mixture. We randomly subsampled the first three components such that the resulting mixture is approximately balanced (albeit with a much smaller sample size due to the subsampling, resulted in a reduced intersection for all methods compared to the original task). As shown in the Table, in this case MixIRLS performs comparably to other methods. This result advocates the effectiveness of MixIRLS in imbalanced mixtures.

2. **Component separation in simulations** [in response to Reviewer LCfo, 16iH]. As raised by Reviewer 16iH, in our manuscript we did not examine how well the MLR methods perform under different separation levels. The separation is defined as the distance between the components, $\\|\beta^*_i-\beta^*_j\\|$ for $i\neq j$; in case of more than two components ($K>2$) we consider the average separation level. A-priori, we expect to see a small error in two extreme cases: when the separation is very small, because the mixture is close to being degenerate (with the smallest error at $\\|\Delta\\|=0$, which corresponds to a single linear model); and when the separation is large, as the component identification is easier. In between, the error can be larger. We conducted a simulation with several separation levels, and the results are depicted in Fig. 2(left) in the attached PDF. We observed an interesting phenomenon: while both MixIRLS and EM follow the expected behavior, their peak error lies in different values of $\\|\Delta\\|$. At small separation levels, EM has a smaller error than MixIRLS, and at moderate levels it is MixIRLS which has a smaller error. However, as shown in the figure, in all separation levels MixIRLS has a better intersection score (see definition in item 1). The reason is that at smaller separation levels, EM finds an almost perfectly balanced mixture, which is far from the true mixture proportions $p = (0.7,0.2,0.1)$. In contrast, MixIRLS better clusters the samples into components and approximates the mixture proportions well even when the error in $\beta$ is relatively large. We will further elaborate on this matter in the revised manuscript.

3. **Information limit** [in response to Reviewers LCfo, 2aVS, 16iH]. In Lines 97-99 of our manuscript we wrote: "The information limit on the sample size, namely the minimal number of observations n required to recover $\beta^*$ in the absence of noise, is $n_\text{info}=d/\text{min}(p)$. No method can recover the models given fewer samples." We will revise the wording and prove this claim. In linear regression with $d$-dimensional explanatory variables, one must observe at least $d$ samples, otherwise the system is underdetermined. Hence, in MLR, one must observe at least $d$ samples from each component. Since the number of samples associated with the $k$-th component is $n\cdot p_k$, one must have $n\cdot p_k\geq d,\forall k$. In other words, it must hold that $n\cdot p_\text{min}\geq d$. Hence, the information limit, namely the minimal required sample size, is $n_\text{info}=d/p_\text{min}$.

4. **Discussion on finite-sample analysis** [in response to Reviewers LCfo, 1Ro7, 2aVS and 16iH]. Our current work includes only a population setting analysis and lacks a finite-sample one. While performing a finite-sample analysis is currently beyond our reach, we wish to discuss why it is so. A main challenge revolves around the sequential nature of MixIRLS. Once MixIRLS estimates a component, it removes its associated samples. Analyzing a modified sample set introduces challenging dependencies - the data no longer follow a normal distribution but rather a conditional distribution. In the population setting, various quantities happen to cancel out nicely; see, for example, the derivation of Eq. (D.17) from (D.16). Under the setting of finite sample size, we need to bound these quantities under a conditional distribution. This is in contrast to simultaneous approaches, such as EM, which don't have these dependencies as the sample set is fixed throughout the iterations. We hope to find a way to analyze these quantities in the future.

---

> ### Comment · Reviewer_LCfo · 2023-08-15
> **Measuring clustering performance in the CO2 experiment**
>
> Thanks to the authors for adding the new CO2 experiment. I think having such a concrete application significantly strengthens the paper and I intend to raise my score due to this addition. However, I'm unsure whether the intersection score used to measure performance in this experiment is appropriate.
>
> Specifically, for imbalanced mixtures, the intersection between the labels found by each method and the true labels is not a very informative metric, as it places little weight on rare classes (much as accuracy is not an informative measure for imbalanced classification). On this dataset, a method could conceivably have 92% intersection score without ever identifying the rare (diesel and ethanol) classes, and, from the reported results, it's unclear which, if any, of the methods actually recover these rare classes. So, I suggest instead reporting class-wise performance measures (e.g., recall) separately for each class. The overall intersection score might still be appropriate for selecting the best permutation of the classes, though.
>
> If it turns out that MixIRLS is better able to recover the rare classes, then I think the results are especially impressive and could be better highlighted by a more informative metric. On the other hand, if it turns out that the stronger performance of MixIRLS is mostly due to better distinguishing the two prevalent (regular/premium gasoline) classes, then the conclusion is still positive, but some more careful interpretation/discussion of the results might be in order (e.g., perhaps this is better explained in terms of robustness than in terms of class imbalance).

---

> > ### Author Response · Authors · 2023-08-15
> >
> > We thank the reviewer for the positive assessment of our new CO2 experiment, and for their excellent suggestion for improvement of the performance measure. We agree with the reviewer that the intersection score we used is not sufficiently informative for an imbalanced mixture. Based on a class-wise accuracy measure (please see details below), we arrive at the following conclusion: all methods perform similarly well on the largest class in the data, but Mix-IRLS performs significantly better on the other, smaller classes.
> >
> > Specifically, we used balanced accuracy, defined as the average of sensitivity/recall and specificity. We chose this measure as it is considered suitable for imbalanced mixtures, but other measures yield qualitatively similar results. The best permutation is still chosen according to the overall intersection score. The median results and the errors (mean absolute values) across 50 repetitions of the experiment are presented in the table below.
> >
> > | Class-wise balanced accuracy &nbsp; |     MixIRLS     |      AltMin     |        EM       |        GD       |
> > |:-----------------:|:---------------:|:---------------:|:---------------:|:---------------:|
> > | regular gasoline (49%) | $\textbf{0.58} \pm \textbf{0.00}$ &nbsp; | $\textbf{0.59} \pm 0.03$ &nbsp; | $\textbf{0.56} \pm 0.04$ &nbsp; | $\textbf{0.57} \pm 0.07$ &nbsp; |
> > |premium gasoline (43%) | $\textbf{0.59} \pm\textbf{0.00}$ | $0.48 \pm 0.10$ | $0.48 \pm 0.01$ | $0.40 \pm 0.03$ |
> > | diesel (5%) | $\textbf{0.89} \pm \textbf{0.00}$ | $\textbf{0.78} \pm \textbf{0.13}$ | $0.54 \pm 0.07$ | $0.70 \pm 0.12$ |
> > | ethanol (3%) | $\textbf{0.74} \pm \textbf{0.00}$ | $0.63 \pm 0.07$ | $\textbf{0.77} \pm \textbf{0.08}$ | $0.61 \pm 0.05$ |
> >
> > The results indicate that Mix-IRLS outperforms the other methods on the small classes. EM is the only method that is slightly better than Mix-IRLS on one of the rare classes (ethanol), but on the other rare class (diesel) it is markedly worse. Moreover, importantly, Mix-IRLS is the only method that is stable with respect to the initialization, demonstrating negligible variance across different random initializations. We will include these and related results (such as confusion matrices and additional measures) in our revised manuscript and analyze them in-depth.

---

### Decision · Program_Chairs · 2023-09-21

**Decision:**

Accept (poster)

**Comment:**

All reviewers are in favour of acceptance, although some limitations regarding the theoretical results were noted during the discussion. For example I recommend the authors clarify what, if any, advantages the proposed method has over the MLE.